# Melanocortin 3 receptor regulates hepatic autophagy and systemic adiposity

Tushar P. Patel [1,7], Joo Yun Jun[1,7], Arnold Y. Seo[2], Noah J. Levi [1], Diana M. Elizondo[1], Jocelyn Chen[1], Adrian M. Wong [1], Nicol Tugarinov [1], Elizabeth K. Altman[1], Daniel B. Gehle [1], Sun Min Jung[1], Pooja Patel[1], Mark Ericson[3], Carrie Haskell-Luevano[3], Tamar C. Demby[4], Antony Cougnoux[5], Anna Wolska [6] & Jack A. Yanovski [1] ✉

Systemic lipid homeostasis requires hepatic autophagy, a major cellular program for intracellular fat recycling. Here, we find melanocortin 3 receptor (MC3R) regulates hepatic autophagy in addition to its previously established CNS role in systemic energy partitioning and puberty. Mice with *Mc3r* deficiency develop obesity with hepatic triglyceride accumulation and disrupted hepatocellular autophagosome turnover. Mice with partially inactive human *MC3R* due to obesogenic variants demonstrate similar hepatic autophagic dysfunction. In vitro and in vivo activation of hepatic MC3R upregulates autophagy through LC3II activation, TFEB cytoplasmic-to-nuclear translocation, and subsequent downstream gene activation. MC3R-deficient hepatocytes had blunted autophagosome-lysosome docking and lipid droplet clearance. Finally, the liver-specific rescue of *Mc3r* was sufficient to restore hepatocellular autophagy, improve hepatocyte mitochondrial function and systemic energy expenditures, reduce adipose tissue lipid accumulation, and partially restore body weight in both male and female mice. We thus report a role for MC3R in regulating hepatic autophagy and systemic adiposity.

Proper hepatic autophagy is needed for active energy redistribution, stimulating both the formation and recycling of cellular lipid-droplets (LDs)[1–3]. Through dynamic fat repartitioning, activation of autophagy can prevent obesity-associated complications and liver steatosis in mice and humans[3–5]. Loss of autophagy has been linked to the development of metabolic dysfunction-associated fatty liver disease (MAFLD)[6,7]. Altered hepatic autophagy has been observed previously in obese mouse models[5,8]. Together with MC4R, MC3R is one of the two melanocortin G protein coupled receptors that are mainly

expressed in the central nervous system (CNS) and modulate energy expenditure and systemic adiposity in humans[9–12]. Knockout (KO) mouse studies[13–15] and human association analyses for MC3R variants[16–21] support its role in regulating body weight, adiposity, and puberty[22]. In mice, global *Mc3r* deficiency obesity is a mild, late-onset form of obesity, that increases body weight moderately because it increases fat mass while reducing in lean mass[23,24]. Despite the relatively mild overall obesity phenotype, hepatic triglyceride (TG) accumulation is markedly increased in *Mc3r*[TB/TB]; the floxed transcriptionally

[1]Section on Growth and Obesity, Division of Intramural Research, Eunice Kennedy Shriver National Institute of Child Health and Human Development (NICHD), NIH, Bethesda, MD, USA. [2]Janelia Research Campus, Howard Hughes Medical Institute (HHMI), Ashburn, VA, USA. [3]Department of Medicinal Chemistry, University of Minnesota College of Pharmacy, Minneapolis, MN, USA. [4]Mouse Metabolism Core, National Institute of Diabetes and Digestive and Kidney Diseases, NIH, Bethesda, MD, USA. [5]Section on Molecular Dysmorphology, Eunice Kennedy Shriver National Institute of Child Health and Human Development (NICHD), NIH, Bethesda, MD, USA. [6]Lipoprotein Metabolism Laboratory, Translational Vascular Medicine Branch, National Heart, Lung, and Blood Institute (NHLBI), NIH, Bethesda, MD, USA. [7]These authors contributed equally: Tushar P. Patel, Joo Yun Jun. ✉e-mail: yanovskj@mail.nih.gov

blocked *Mc3r* deficient mouse[23]. Liver microarray analysis, which was confirmed using qPCR, showed more fatty acid synthesis gene upregulation and lower serum TG[23] and reduced fasting induced lipolysis in white adipose tissue[24], driving nutrient partitioning towards liver steatosis and systemic adiposity.

Previous studies have demonstrated that *Mc3r* global knockout mice do not have excessive food intake; rather they are described as being hypophagic relative to controls[13,15,23]. Additionally, some studies show evidence for reduced energy expenditure, respiratory exchange ratio, and locomotion in *Mc3r^TB/TB* mice[23]. However, other studies have demonstrated normophagia and normal global metabolism in *Mc3r* deficiency[24], leaving the mechanism of commonly observed increased adiposity undetermined[14,23,25,26].

Interestingly, unlike other monogenic obesity models with defective hypothalamic signaling (e.g., *Mc4r*KO mice), recent studies suggest *Mc3r*KO mice develop an obese phenotype in part via MC3R insufficiency outside of the CNS[8,23]. Indeed, neither nervous system-specific rescue of *Mc3r* with nestin-cre nor hypothalamic rescue with steroidogenic factor (SF1)-cre fully reverses the abnormal fat mass gain in *Mc3r*KO mice on a normal chow diet, and neither CNS-*Mc3r* recovery model has a significant effect on body weight for mice given a high-fat diet[23]. Regardless of nutritional state, cellular mechanisms for lipid mobilization and nutrient partitioning play important roles in loss of function (LoF) MC3R-associated obesity. In our humanized MC3R knock-in mouse model[27], a coding sequence variant (C17A + G241A) that impairs MC3R activity and appears to bias stem cell differentiation towards more adipogenic phenotypes, seems to cause obesity at least in part through peripheral mechanisms.

The liver has a central role in lipid transport and metabolism, including but not limited to the synthesis and catabolism of lipoproteins. Dietary short-chain fatty acids from the portal circulation, adipocyte-derived fatty acids, and excess triglycerides in chylomicron remnants are all transported to the liver[28]. Intrahepatic fatty acids are then reassembled as triglycerides in the endoplasmic reticulum, and combined with phospholipids, cholesterol, cholesteryl esters, and apoB molecules[29] to form VLDL that is generally released into the circulation, where other tissues will utilize triglycerides for energy or will store them for later use. When VLDL secretion is not well matched with its intrahepatic synthesis, hepatic steatosis results[30].

After identifying that *Mc3r* transcripts are expressed in liver tissue, we sought to examine if liver *Mc3r* is important for hepatic and/or systemic fat accumulation in *Mc3r* deficiency, and if so, to investigate the underlying mechanisms. Given that proper hepatic autophagy is needed for active energy redistribution, stimulating both formation and recycling of cellular LDs[1–3] we aimed to investigate the liver-specific role of *Mc3r* in LD-autophagy, finding that activation of MC3R signaling induces autophagy in wild-type but not Mc3r-deficient mice and that autophagy flux is dysregulated in *Mc3r* deficiency, with defective lysosomal turnover. Further, our data show liver-specific *Mc3r* rescue in the context of global *Mc3r* deficiency restored liver autophagy and reduced adiposity due to changes in metabolic processes affecting energy expenditure.

## Results

### *Mc3r* expression regulates systemic and hepatic adiposity

Prior work using RNAseq approaches indicated tissue-specific regulation of *Mc3r* expression in both mouse[31,32] and axolotl liver[33]. Although transcriptome databases reported no *Mc3r* transcripts in mouse liver, we examined *Mc3r* transcript levels from control and *Mc3r^TB/TB* mice (MC3R-deficient due to insertion of a transcription blocker) using the more sensitive droplet digital PCR (ddPCR) method[34], and found murine hepatic *Mc3r* expression (Fig. 1a). Although the estimated total *Mc3r* transcript level was lower in liver compared to hypothalamus (Fig. 1b), it was significantly higher in livers from control mice compared to those of *Mc3r^TB/TB* mouse livers at Zeitgeber time (ZT) 12 and

not at ZT2 (Fig. 1a). Notably, we found that expression of *Mc3r* transcripts in the control liver (Fig. 1a) was high at nighttime (ZT12), when MC3R ligands circulate in rodent plasma[35,36].

To understand *Mc3r*'s peripheral functions during obesogenesis, we began by examining systemic and hepatic adiposity in mouse genetic models with either no *Mc3r* activity (*Mc3r^TB/TB*) or reduced *Mc3r* activity due to knock-in of a human mutated *MC3R* (*MC3R^hDM/hDM*)[27] that is linked to childhood and adult obesity[18,37]. With regular chow diet, we found progressive weight gain (Fig. 1c) in *Mc3r^TB/TB* and *MC3R^hDM/hDM* mice, with significantly increased total adiposity (Fig. 1d) and liver TG accumulation (Fig. 1e) by 12 weeks of age, compared to control wild-type mice (i.e., *Mc3r^+/+*, C57BL/6) or knock-in mice expressing the intact human MC3R (i.e., *MC3R^hWT/hWT*). Oil Red O staining from *Mc3r^TB/TB* and *MC3R^hDM/hDM* mice confirmed elevated hepatic fat deposition (Fig. 1f). The elevated liver TG of *Mc3r* insufficiency was associated with increased LD deposition, as detected by transmission electron microscopy (TEM) in liver cells in the fed condition (Fig. 1g, h). Since no significant differences were seen in liver or systemic fat development between *Mc3r^+/+* and *MC3R^hWT/hWT* mice (Fig. 1), we refer to both such mice as controls. Our control mice, *Mc3r^+/+* and *MC3R^hWT/hWT* showed no significant differences in liver or systemic fat development (Fig. 1).

### Altered hepatic lipid droplet recycling and autophagy in MC3R-insufficient mice

Under chow diet, both *Mc3r^TB/TB* and *MC3R^hDM/hDM* livers displayed significantly increased TG levels compared to the levels seen in the control livers (Fig. 1e), suggesting potential defects in fat metabolism in the livers due to defective MC3R activity. Previously, we found hepatic TG levels were not different in mice with defective *Mc3r* compared to the control mice during high-fat diet[27]. We examined liver LD quantity (the major hepatocyte TG storage) from chow diet fed *Mc3r^TB/TB* and *MC3R^hDM/hDM* mice after overnight (O/N) fasting, which normally stimulates hepatic fat uptake and TG production. With TEM, we found that a similar number of LDs formed in *Mc3r* defective livers upon O/N fasting (Supplementary Fig. 1a, b), confirming there was no defect in hepatic fatty acid uptake or TG deposition in *Mc3r*-insufficient mice.

Next, due to liver autophagy's essential role in hepatic LD recycling and fat catabolism[1,3,38], we aimed to determine if defective *Mc3r* leads to TG accumulation by impairing LD recycling and degradation by liver autophagy. Using TEM to examine autophagosome (AP) organization, we found similar AP production in the control versus *Mc3r^TB/TB* mice after O/N fasting (Fig. 2a, b), suggesting the capability for AP biogenesis itself is unchanged in *Mc3r* defective livers. However, unlike AP from the control cells that contained electron-dense structures that likely are degradative debris of intracellular components, many AP from *Mc3r^TB/TB* or *MC3R^hDM/hDM* mice contained intact membrane structures (Fig. 2a and c, see arrows and arrowheads indicating mitochondrial outer and inner membranes). This suggests inefficient AP degradation in the *Mc3r* defective livers. In addition, liver TEM images from *Mc3r^TB/TB* or *MC3R^hDM/hDM* mice often showed multiple AP structures in very close proximity, unlike the discrete and isolated AP structures observed in control cells (Fig. 2a and c, asterisks). These AP aggregates with undegraded contents in the livers of *Mc3r^TB/TB* or *MC3R^hDM/hDM* mice led us to hypothesize that excessive hepatic TG accumulation in MC3R defective mice results from compromised hepatic TG degradation. Thus, we studied if disrupting the hepatic *Mc3r* pathway dysregulates the autophagic recycling process.

### Defective autophagosome turnover in MC3R deficient liver

Impaired lysosomal AP clearance could accumulate APs with undegraded subcellular compartments. To examine the liver AP recycling program, we monitored LC3II expression in the livers of *MC3R^hDM/hDM* and *Mc3r^TB/TB* mice. Since LC3II is converted from LC3I by lipidation during autophagy induction and constantly undergoes recycling after autophagolysosome formation, the total level of

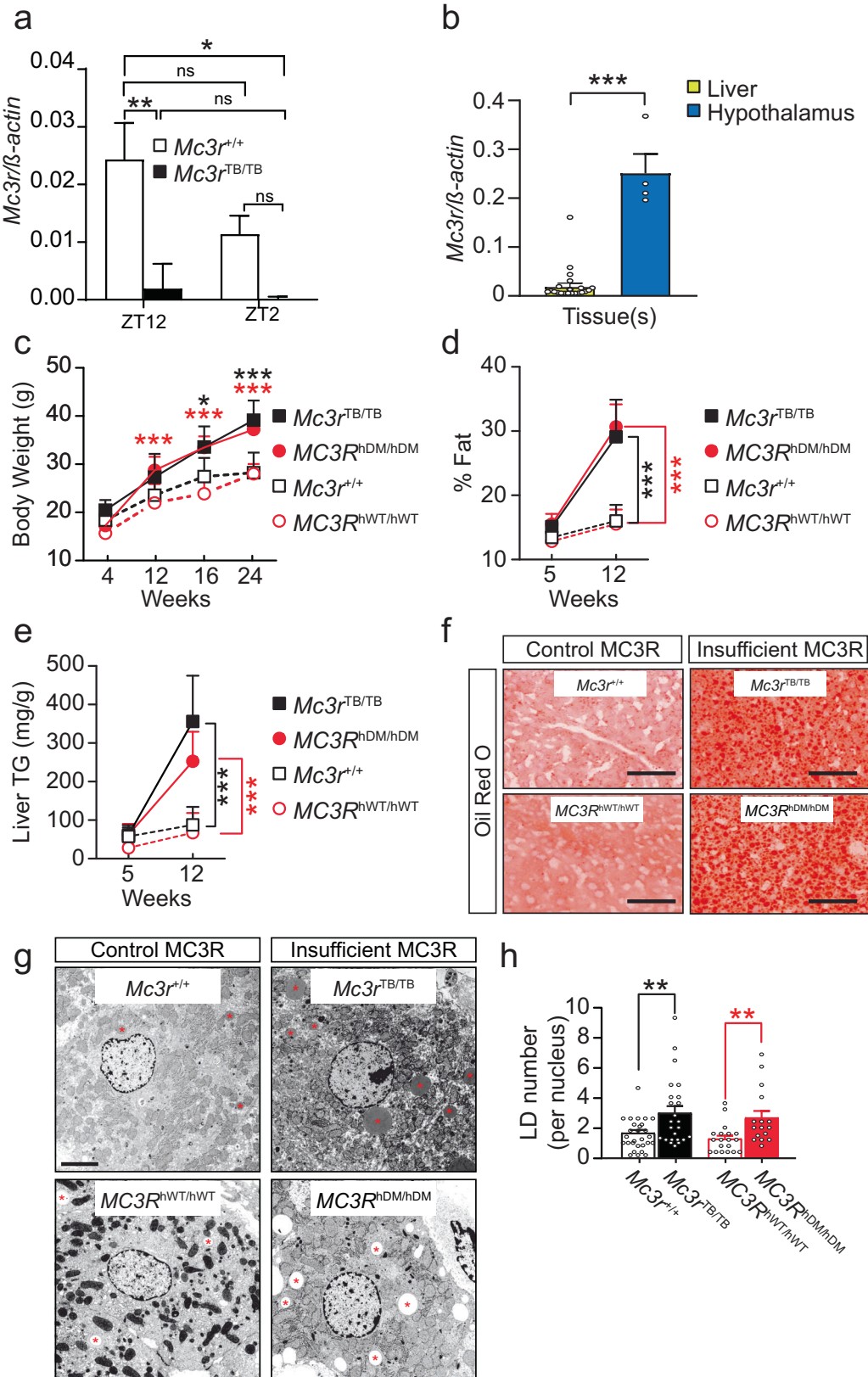

cellular LC3II (normalized to beta-actin) was examined as an indicator of overall cellular autophagy activity. We found that under fed conditions (when mice maintain basal autophagy activity in the liver), both *Mc3r^TB/TB* and *MC3R^hDM/hDM* mice exhibited much greater basal LC3II protein levels in liver compared to control mice (Fig. 2d, e) due to defective *Mc3r*, leading to LC3II accumulation. As

expected, under fasting conditions, there was significant induction of liver LC3II levels in control mice (Fig. 2d, e), confirming our fasting condition enhances hepatic autophagy. However, the similarly fasted *Mc3r^TB/TB* and *MC3R^hDM/hDM* mice failed to increase liver LC3II (Fig. 2d, e), suggesting impaired autophagic regulation in *Mc3r*-deficient livers.

**Fig. 1 | *Mc3r* insufficiency increases systemic and hepatic adiposity in mice.**
**a** Normalized droplet digital PCR *Mc3r* transcript levels from *Mc3r*$^{+/+}$ liver ($n = 37$) at ZT12 (6:00 PM) and from *Mc3r*$^{+/+}$ liver ($n = 35$) at ZT2 (8:00 AM) and *Mc3r*$^{TB/TB}$ liver ($n = 18$) at ZT12, and from *Mc3r*$^{TB/TB}$ liver ($n = 19$) at ZT2. **b** Normalized *Mc3r* transcript levels from *Mc3r*$^{+/+}$ liver ($n = 21$) and hypothalamus ($n = 4$, with each RNA extraction sample combining 6 pooled hypothalamus isolations); **c** Body weight measured from *Mc3r*$^{+/+}$ ($n = 8$), *Mc3r*$^{TB/TB}$ ($n = 10$), *MC3R*$^{hWT/hWT}$ ($n = 12$) and *MC3R*$^{hDM/hDM}$ ($n = 11$) independent observations. **d** Total body % fat from *Mc3r*$^{+/+}$ ($n = 9$), *Mc3r*$^{TB/TB}$ ($n = 8$), *MC3R*$^{hWT/hWT}$ ($n = 7$) and *MC3R*$^{hDM/hDM}$ ($n = 7$) independent observations. **e** Liver triglycerides (TG) from *Mc3r*$^{+/+}$ ($n = 6$), *Mc3r*$^{TB/TB}$ ($n = 5$), *MC3R*$^{hWT/hWT}$ ($n = 7$) and *MC3R*$^{hDM/hDM}$ ($n = 6$) independent observations. **f** Oil Red O

images of livers ($n = 2$/genotype from two independent observations)
**g** Transmission Electron Microscope (TEM) images of livers from mice in the fed state ($n = 1$–2/genotype from two independent experiments) at 8–12 weeks of age (**h**) Quantification of lipid droplet (LD) number from TEM images ($n = 16$–28 *images*/genotype from two independent observations) shown in (**g**). Asterisks indicate LD. Data are represented as mean ± SEM. Groups were compared by one-way ANOVA followed by Tukey's HSD (**a**), paired two-tailed Student's *t*-test (**b**, **h**), two-way ANOVA followed by Tukey's HSD (**c**–**e**). * $p < 0.05$; ** $p < 0.01$; *** $p < 0.001$; male mice (8–12 weeks of age) under chow-fed diet were used unless otherwise indicated. Scale bar, 50 µm (**f**) and 5 µm (**g**). Source data are provided as a Source data file.

To examine which functions of liver autophagy are dysregulated in *Mc3r*$^{TB/TB}$ and *MC3R*$^{hDM/hDM}$ mice, we studied AP turnover kinetics (AP flux) in primary hepatocytes by using chloroquine, a well-known AP-lysosome fusion blocker[39]. To minimize pleiotropic effects, we designed an acute AP flux assay by treating with chloroquine for only 30 min[39]. Blockade of AP degradation by chloroquine marginally increased LC3II in the control hepatocytes under fed or serum starved conditions without showing significant cytotoxic phenotypes (Fig. 2f and Supplementary Fig. 1c). Our estimated AP flux data suggested that under serum starvation, 10 to 20% of AP undergo lysosomal digestion in control hepatocytes (Fig. 2g). In contrast, the AP flux was perturbed in the similarly treated cells from *Mc3r*$^{TB/TB}$ or *MC3R*$^{hDM/hDM}$ mice (Fig. 2f, g). Furthermore, supporting dysregulated AP degradation in the hepatocytes with defective *Mc3r*, we also observed accumulated p62/SQSTM1 (a key cargo protein that normally undergoes autophagic degradation[40]) during serum starvation in Mc3r-insufficient hepatocytes (Fig. 2f and Supplementary Fig. 1c). As an obesity control, we also tested autophagy flux in *Mc4r*$^{+/-}$ mice, as they display a comparable adiposity with *Mc3r*$^{TB/TB}$ mice, though not as severe an obesity phenotype as is seen in *Mc4r*$^{-/-}$ mice[13,41,42]. Importantly, upon serum starvation, hepatocytes isolated from *Mc4r*$^{+/-}$ mice could maintain AP flux to a similar extent as seen in control hepatocytes, as evidenced by normal activity of p62 degradation (Fig. 2f, g). The dysregulated autophagy observed in *Mc3r*$^{TB/TB}$ and *MC3R*$^{hDM/hDM}$ liver is thus not due solely to the obesity phenotype, rather these data indicate a distinct role of *Mc3r* in autophagy regulation.

To further monitor hepatic autophagy processes, we generated transgenic GFP-LC3 mice with *Mc3r* insufficiency. We found that hepatocytes carrying intact *Mc3r* displayed dispersed cytoplasmic GFP-LC3 with very few discrete LC3 vesicle structures during fed conditions (Fig. 2h). Conversely, hepatocytes from *Mc3r*$^{TB/TB}$ or *MC3R*$^{hDM/hDM}$ mice displayed many GFP-LC3 vesicles that were aggregated even without serum starvation (Fig. 2h, Inlay). Similar LC3 aggregates were formed in control hepatocytes when AP degradation was blocked by chloroquine (Supplementary Fig. 1d). Additional 3D reconstruction of GFP-LC3 structures further revealed augmented AP volumes in *Mc3r*-insufficient hepatocytes compared to control cells (Fig. 2i). These results clearly support impaired AP turnover caused by *Mc3r* deficiency in mouse liver.

## Liver-specific role of MC3R in controlling adiposity and glucose metabolism

To interrogate potential hepatic functions of *Mc3r*, we selectively re-expressed liver *Mc3r* in *Mc3r*$^{TB/TB}$ mice by crossing them with mice expressing Cre-recombinase only in hepatocytes (Supplementary Fig. 2a). Using the resulting progeny (*Mc3r*$^{Hep/Hep}$ mice expressing *Mc3r* only in the liver), we found liver-specific *Mc3r* recovery significantly improved percentage of fat and lean mass relative to *Mc3r*$^{TB/TB}$ but lean mass percentage was still significantly reduced, and fat mass percentage still significantly increased, relative to control mice (Fig. 3a, b). The abnormalities of *Mc3r*$^{TB/TB}$ mice in serum insulin (Fig. 3c) and fasting glucose concentrations (Fig. 3d), and consequently in insulin sensitivity by glucose and insulin tolerance test area under curve (AUC)

analyses (Fig. 3f, h) were also ameliorated in *Mc3r*$^{Hep/Hep}$ mice. Restoring *Mc3r* expression only in the liver led to partially improved overall body weight regulation during chow diet compared to the *Mc3r* null mice in both male (Fig. 3i) and female (Fig. 3j) mice. We also compared weight gains in contemporaneously studied *Mc4r*$^{+/-}$ mice, the far less severely affected heterozygotes of the severe monogenetic obesity *Mc4r*$^{-/-}$ mouse (Supplementary Fig. 3a, b). Our data finding *Mc3r*$^{TB/TB}$ homozygotes gain similar body weight compared to *Mc4r* heterozygotes confirm that *Mc3r* insufficiency is a moderate form of obesity that can be, to a certain extent, improved by restoring hepatic *Mc3r* expression.

## Liver-specific role of MC3R in controlling hepatic fat accumulation and fatty acid metabolism

To examine tissue fat accumulation and histology in liver-specific *Mc3r* recovery, we observed elevated hepatic fat deposition in *Mc3r*$^{TB/TB}$ that was reversed in the *Mc3r*$^{Hep/hep}$ mice as assessed by H&E staining (Fig. 4a) and Oil Red O staining (Fig. 4b). Tissue TG content, markedly increased in *Mc3r*$^{TB/TB}$ was returned towards normal in *Mc3r*$^{Hep/Hep}$, suggesting restored hepatic lipid homeostasis (Fig. 4c). Importantly, hepatic *Mc3r* reactivation fully reversed the increased liver weight (Fig. 4d), but only partially reduced epididymal white adipose tissue (eWAT) weight (Fig. 4e) compared to *Mc3r*$^{TB/TB}$ and control mice.

To further evaluate the impact of reactivating hepatic *Mc3r* on circulatory lipids at the systemic level, we measured serum triglyceride concentrations, which were elevated in *Mc3r*$^{TB/TB}$ mice and restored to values not different from those of *Mc3r*$^{+/+}$ in *Mc3r*$^{Hep/Hep}$ (Fig. 4f). Serum non-esterified fatty acid (NEFA) concentrations were significantly higher in *Mc3r* deficient mice; hepatic *Mc3r* recovery similarly normalized NEFA (Fig. 4g) without a significant change in serum glycerol concentrations (Fig. 4h), suggesting improving insulin sensitivity and hepatic fatty acid metabolism. Elevated serum total cholesterol was also normalized in *Mc3r*$^{Hep/hep}$ (Fig. 4i), which is consistent with the observed improvements in systemic obesity and liver steatosis.

We also examined hepatic mRNA expression of fatty acid metabolic genes (Fig. 4j). *Cd36*, the primary transporter responsible for uptake prior to fatty acid esterification process, was significantly increased in *Mc3r*$^{TB/TB}$ mice in the fasted state, though not in the fed state. *Mc3r*$^{Hep/Hep}$ hepatic recovery mice showed significantly reduced *Cd36* in the fasted state compared to *Mc3r*$^{TB/TB}$ with values similar to control mice (Fig. 4f). These findings suggest that *Mc3r* deficiency leads to an overall increase in total fatty acid esterification. Liver lipid droplet surface binding protein *Cidec* (cell death inducing DFFA like effector C), was found to be upregulated in *Mc3r*$^{TB/TB}$, as described previously[23], consistent with the excessive lipid droplet formation in *Mc3r*$^{TB/TB}$. *Mc3r*$^{Hep/Hep}$ returned *Cidec* to control levels compared to *Mc3r*$^{TB/TB}$, suggesting improvement in insulin sensitivity. In both wild-type control and hepatic recovery groups, the liver showed increased *Cidec* expression levels when mice were fasted. At the same time, *Mc3r* global deficiency did not respond to fasting and exhibited lower *Cidec* expression compared to the fed state, consistent with a defect in lipid droplet formation and lipid droplet autophagy. We observed no significant differences between *Mc3r*$^{TB/TB}$ and controls for hepatic expression of the key rate-limiting enzymes of lipogenesis (fatty acid

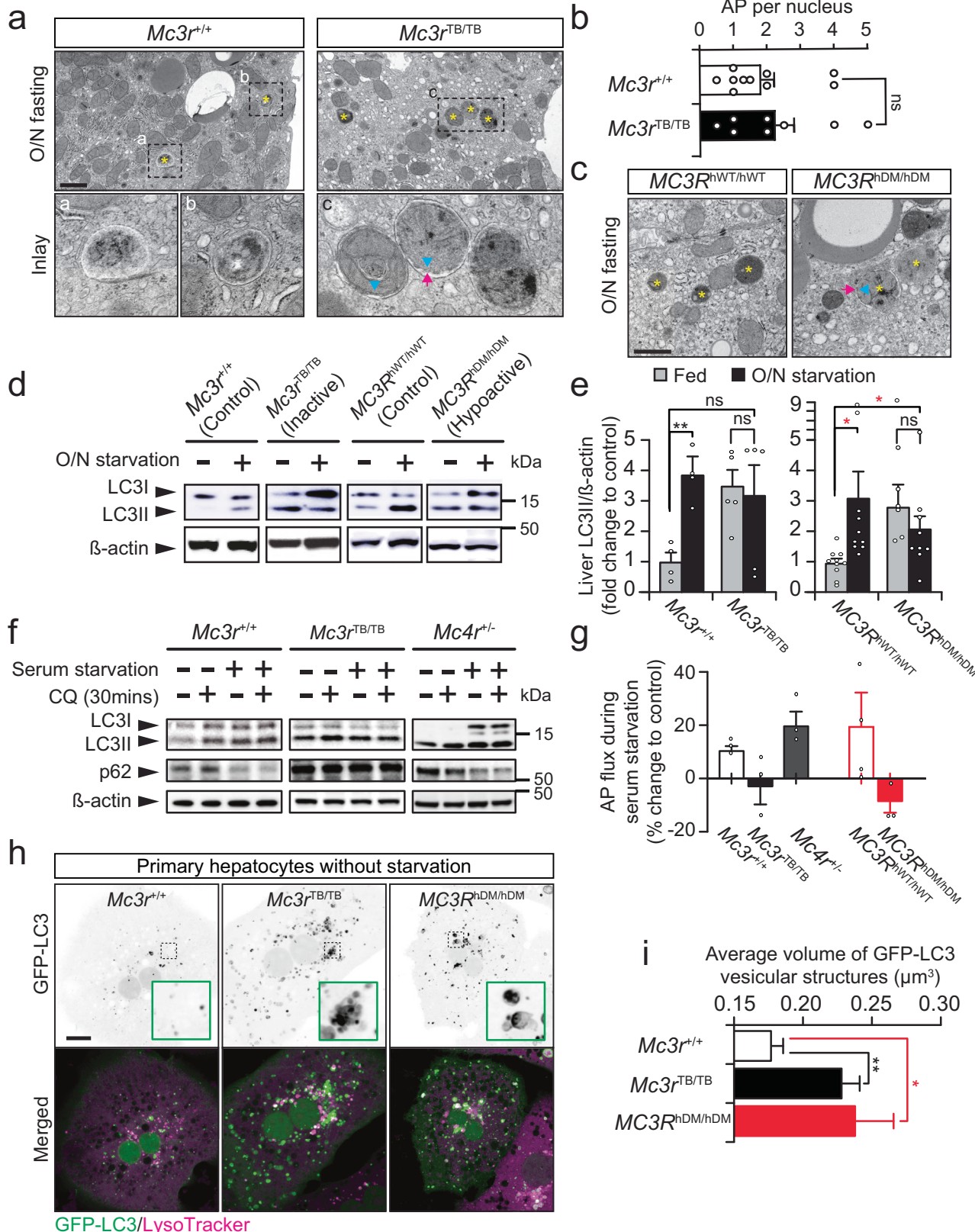

synthase; *Fasn*) and of Lipolysis (*Lipe*). In the fasted state, hepatic *Ppargc1a* was notably lower in *Mc3r*^TB/TB than control mice but returned to a normal level in the hepatic recovery mice. This suggests that *Mc3r* deficiency may inhibit the oxidation of fatty acids, which in turn affects the breakdown of excess fatty acids. The reversal of a significant systemic adiposity phenotype could potentially be explained by the altered regulation of hepatic fat accumulation and fatty acid metabolism in *Mc3r*^Hep/Hep.

**Effect of MC3R deficiency in regulating feeding behavior**

We monitored food intake for 15 days in single-housed mice in our vivarium, which was not significantly different in *Mc3r*^TB/TB and *Mc3r*^Hep/Hep

**Fig. 2 | Altered hepatic autophagy and LD recycling in *Mc3r* insufficient mice.**
**a** Transmission Electron Microscope images showing liver autophagosomes (AP) from overnight (O/N) fasted *Mc3r*[+/+] and *Mc3r*[TB/TB] (*n* = 1–2/genotype from two independent experiments). **b** Quantification of AP number from TEM images from both *Mc3r*[+/+] (*n* = 15–16 images/genotype from two independent observations) shown in (**a**). **c** TEM images showing liver AP from overnight (O/N) fasted *Mc3r*[hWT/hWT] and *MC3R*[hDM/hDM]. **d** Western blot images of liver lysates from *Mc3r*[+/+], *Mc3r*[TB/TB], *Mc3r*[hWT/hWT] and *MC3R*[hDM/hDM]. **e** Quantification of LC3II normalized by actin from both fed-fasted *Mc3r*[+/+] (*n* = 4) independent observations, *Mc3r*[TB/TB] (*n* = 5) independent observations, *MC3R*[hWT/hWT] (*n* = 10) independent observations, and *MC3R*[hDM/hDM] (*n* = 9 (fed) and 10 (fasted) independent observations) in (**d**). **f** Western blot images of primary hepatocytes from *Mc3r*[+/+], *Mc3r*[TB/TB] and *Mc4r*[+/−] after 1 h serum starvation or 30 min of 10 mM chloroquine (CQ) treatment. **g** Quantification of AP flux estimated in (**f**) and Supplementary Fig. 1c from *Mc3r*[+/+] (*n* = 3), *Mc3r*[TB/TB] (*n* = 3), *MC3R*[hWT/hWT] (*n* = 4) and *MC3R*[hDM/hDM] (*n* = 3). **h** Fluorescent images of primary hepatocytes from *Mc3r*[+/+], *Mc3r*[TB/TB] and *MC3R*[hDM/hDM] transgenic mice carrying GFP-LC3 after staining with Lysotracker Red. **i** Quantification of AP volume in (**h**) (*n* = -1500/2 mice/genotype from two independent experiments). Asterisks indicate AP. Arrows and arrowheads indicate mitochondrial outer and inner membranes, respectively. Data are represented as mean ± SEM. Groups were compared by unpaired two-tailed Student's *t*-test (**b**), ordinary one-way ANOVA followed by Kruskal–Wallis test followed by Dunn's multiple comparisons test or Tukey's HSD test (**e**, **g**, **i**). * $p < 0.05$; ** $p < 0.01$; *** $p < 0.001$, Scale bar, 1 μm (**a**, **c**) and 10 μm (**h**). Source data are provided as a Source data file.

---

mice compared to *Mc3r*[+/+] (Supplementary Fig. 3c), even after adjusting for body weight or lean mass. Additionally, feeding efficiency (FE) did not differ between groups (Supplementary Fig. 3d). However, when food intake monitoring was conducted using CLAMS metabolic chambers (Fig. 5a–c), food intake differences emerged. Daily energy intake was significantly reduced in *Mc3r*[TB/TB] and *Mc3r*[Hep/Hep] mice compared to *Mc3r*[+/+] at 22 °C (Fig. 5b). After adjusting energy intake for body weight, lean & fat mass, or lean mass, *Mc3r*[TB/TB] and *Mc3r*[Hep/Hep] remained significantly lower than *Mc3r*[+/+] (Supplementary Table 1). At thermoneutrality (30 °C), energy intake was significantly reduced in *Mc3r*[TB/TB] vs. *Mc3r*[+/+] only during the dark phase (Fig. 5c). This reduction in energy intake is consistent with previously reported findings that global transcriptionally blocked *Mc3r*-deficient mice are relatively hypophagic[23,25]. Here, re-expression of *Mc3r* only in the liver did not restore energy intake, suggesting that feeding behavior alterations in *Mc3r* deficiency are centrally (CNS) regulated and not via hepatic regulation[23].

## Effect of hepatic *Mc3r* reactivation on energy expenditure, locomotion, and brown adipose tissue thermogenesis

To examine whether differences in total energy expenditure (TEE) contribute to the altered energy balance seen in *Mc3r*[HEP/HEP], we performed indirect calorimetry at ambient 22 °C and thermoneutral temperature 30 °C during chow feeding using *Mc3r*[+/+], *Mc3r*[TB/TB], and *Mc3r*[Hep/Hep] mice (Fig. 5d). TEE was significantly reduced in *Mc3r*[TB/TB] mice compared to *Mc3r*[+/+] in both light and dark phases. Notably, with *Mc3r* reactivation, *Mc3r*[Hep/Hep] mice exhibited a significant increase in TEE compared to *Mc3r*[TB/TB] in both light and dark phases, which also translated to total 24-h energy expenditure (EE) at ambient 22 °C (Fig. 5e, f). For TEE at a thermoneutral temperature of 30 °C, *Mc3r*[TB/TB] was significantly lower compared to both *Mc3r*[+/+] and *Mc3r*[Hep/Hep] groups in the light phase and for total 24-h EE (Fig. 5e, f). Recovery from reduced 24-h total energy expenditure in the *Mc3r*[TB/TB] phenotype indicates that the role of hepatic *Mc3r* in autophagy is a key mechanism involves in lipid metabolism, energy expenditure and body weight. For *Mc3r*[TB/TB] mice, TEE was also reduced significantly compared to both *Mc3r*[+/+] or *Mc3r*[Hep/Hep] groups after adjusting for body weight and lean & fat mass at both temperature conditions (Supplementary Table 1). At a thermoneutral temperature of 30 °C, total energy expenditure (TEE), adjusted for body weight and lean and fat mass, showed a trend for an intermediate phenotype in *Mc3r*[Hep/Hep]; however, it was not statistically significant compared to *Mc3r*[TB/TB].

Compared to control mice (Fig. 5g–i), total oxygen consumption (VO₂) was significantly reduced in *Mc3r*[TB/TB] mice, but not in *Mc3r*[Hep/Hep] mice, at ambient 22 °C and 30 °C. Total VO₂ was significantly increased in *Mc3r*[Hep/Hep] versus *Mc3r*[TB/TB] mice at ambient 22 °C (Fig. 5g–i). After adjusting VO₂ for body weight, and total lean & fat mass, Mc3r[TB/TB] also significantly differed from *Mc3r*[+/+] at both temperatures. At a thermoneutral temperature of 30 °C, *Mc3r*[Hep/Hep] mice's total VO₂ exhibited a significant recovery relative to *Mc3r*[TB/TB], even after adjustment for body weight and total lean & fat mass, confirming the significant

recovery phenotype (Supplementary Table 1). This difference in VO₂ consumption is a key indicator of improvement in metabolic activity.

Respiratory exchange ratio (RER) appeared to show lower total RER in *Mc3r*[TB/TB] compared to *Mc3r*[+/+], while *Mc3r*[Hep/Hep] mice were significantly increased compared to *Mc3r*[TB/TB] mice at 22 °C in the dark phase and 24-h RER at 30 °C (Fig. 5j–l). Restored RER in *Mc3r*[Hep/Hep] implies an altered substrate preference (from fat to carbohydrates) for generating energy, as might be expected in a mouse with less adipose tissue.

We also monitored locomotion activity by beam breaks in the metabolic cages during the light and dark phase cycles at 22 °C and 30 °C using *Mc3r*[+/+], *Mc3r*[TB/TB], and *Mc3r*[Hep/Hep] mice. Total activity and ambulatory activity (physical activity) were all similar between groups during both light and dark phases (Supplementary Fig. 4a–f), suggesting that locomotion (by beam breaks) is not a contributing factor in the change in the obesity phenotype of *Mc3r* deficiency.

We checked brown adipose tissue (BAT) UCP-1 activity to monitor thermogenesis after studying mice either at ambient 22 °C or after cold exposure (6 °C) for 6 h. UCP-1 staining increased after cold exposure compared to ambient temperature in all groups without notable differences in the signal intensity between *Mc3r*[+/+], *Mc3r*[TB/TB], and *Mc3r*[Hep/Hep] in either temperature condition (Supplementary Fig. 4g). Immunoblotting results for UCP-1 protein expression were also similar between the groups (Supplementary Fig. 4h, i). *Mc3r* deficiency did not significantly alter BAT UCP-1 activity or brown adipose tissue thermogenesis, one of the alternative mechanisms for dissipating energy as heat.

## Hepatic *Mc3r* reactivation restores adaptive energy expenditure, cellular respiration, and lipid droplet autophagy

To further investigate the critical role of hepatic MC3R in energy balance, we studied how *Mc3r* deficiency affects adaptive energy expenditure in response to cold exposure, when the body requires energy for thermogenesis. Adaptive energy expenditure (EE) in response to cold exposure was significantly blunted in *Mc3r*[TB/TB] compared to *Mc3r*[+/+] mice, even within the first few hours (Fig. 6a, b). In contrast, *Mc3r*[Hep/Hep] mice normally responded to cold exposure EE compared to *Mc3r* deficient mice after 6 h at 6 °C (Fig. 6a, b). Corroborative results were found from oxygen consumption measurements. Oxygen consumption (VO₂) was significantly reduced in *Mc3r*[TB/TB] mice and significantly different from *Mc3r*[+/+] and *Mc3r*[Hep/Hep] at 6 °C (Fig. 6c, d). These results suggest that hepatic *Mc3r* is essential for regulating adaptive energy expenditure in response to thermogenic energy demands.

To elaborate the mechanism for altered energy expenditure at the hepatocyte level, we cultured primary hepatocytes to study cellular mitochondrial respiration using the Seahorse Mito Stress Test. Compared to *Mc3r*[+/+] and *Mc3r*[Hep/Hep] group, *Mc3r*[TB/TB] cells showed reduced basal level respiration (Fig. 6e). We observed restoration in basal respiration, measured by the oxygen consumption rate, in *Mc3r*[Hep/Hep] hepatocytes compared to the *Mc3r*[TB/TB] group (Fig. 6f). Injection of the mitochondrial uncoupler FCCP allows a measurement of maximum

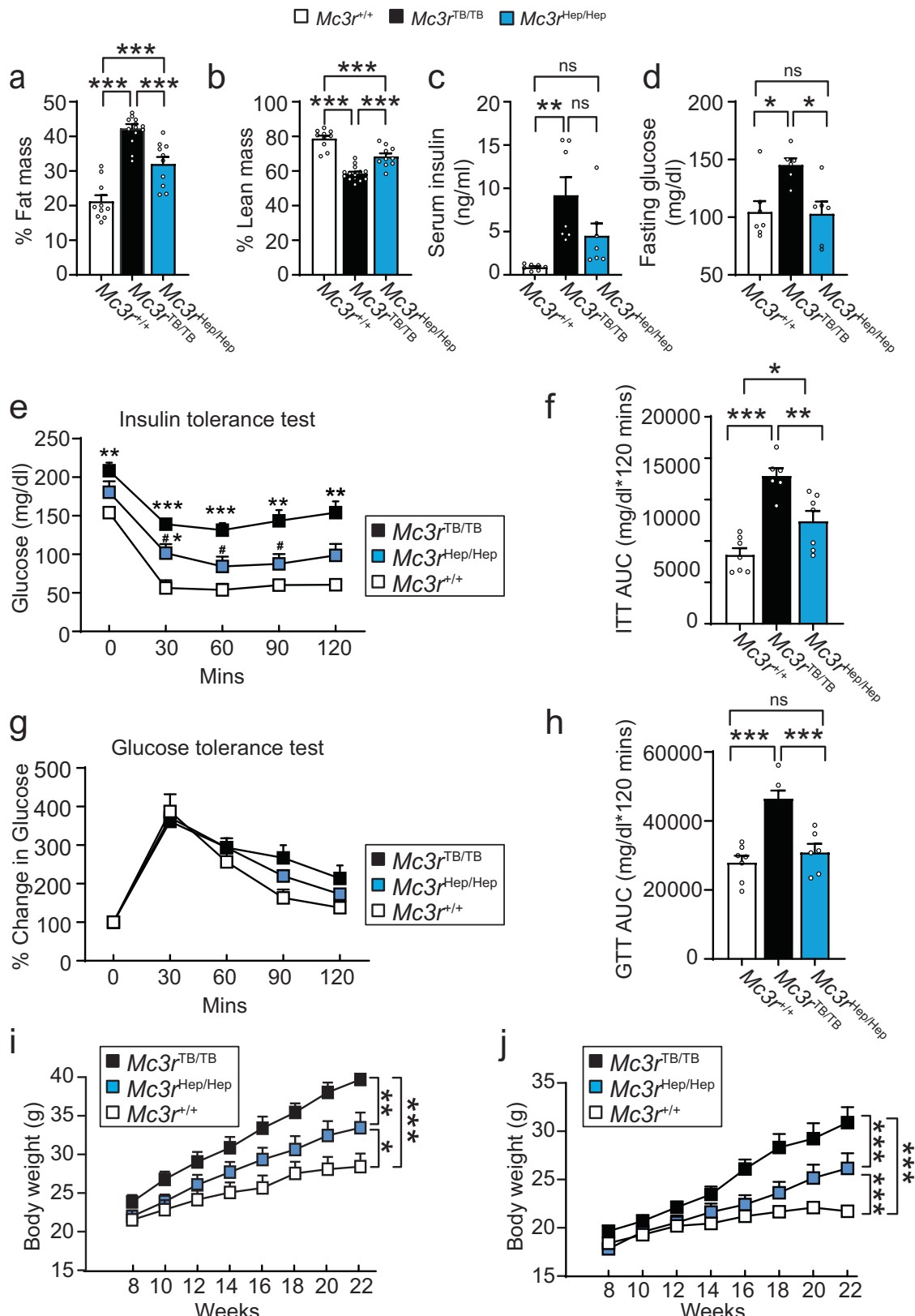

respiration which reflects mitochondrial ability to respond to increased energy demands (Fig. 6e). Maximum respiration was also restored to normal in $Mc3r^{Hep/Hep}$, which was significantly greater than $Mc3r^{TB/TB}$ group and similar to the $Mc3r^{+/+}$ control group (Fig. 6g). These Seahorse assay results in hepatocytes align with the energy expenditure data observed in the mouse models, further validating the

importance of liver $Mc3r$ for energy regulation. These findings strongly support the conclusion that hepatic $Mc3r$ plays a crucial role in adapting to external stimuli, contributing significantly to systemic obesity through its effects on peripheral tissues.

To understand the role of $Mc3r$ using an unbiased approach, bulk RNA-sequencing analysis was performed on liver samples from $Mc3r^{+/+}$,

**Fig. 3 | Hepatic *Mc3r* reactivation restores body weight and body composition in *Mc3r* insufficient mice. a** % Fat mass from *Mc3r*^+/+^ (*n* = 10), *Mc3r*^TB/TB^ (*n* = 14) and *Mc3r*^Hep/Hep^ (*n* = 10) independent observations. **b** % Lean mass from *Mc3r*^+/+^ (*n* = 10), *Mc3r*^TB/TB^ (*n* = 14) and *Mc3r*^Hep/Hep^ (*n* = 10) independent observations. **c** Fasting serum insulin from *Mc3r*^+/+^, *Mc3r*^TB/TB^ and *Mc3r*^Hep/Hep^ (*n* = 7) independent observations/genotype. **d** Fasting glucose from *Mc3r*^+/+^ (*n* = 7), *Mc3r*^TB/TB^ (*n* = 6) and *Mc3r*^Hep/Hep^ (*n* = 6) independent observations. **e** Insulin tolerance test (ITT) measured from *Mc3r*^+/+^ (*n* = 7), *Mc3r*^TB/TB^ (*n* = 6) and *Mc3r*^Hep/Hep^ (*n* = 7) independent observations. * *p* < 0.05; ** *p* < 0.01; *** *p* < 0.001 group compared to *Mc3r*^+/+^ or #*p* < 0.05 group compared to *Mc3r*^TB/TB^. **f** ITT AUC measured from (**e**). **g** Glucose tolerance test (GTT) measured from *Mc3r*^+/+^ (*n* = 7), *Mc3r*^TB/TB^ (*n* = 6) and *Mc3r*^Hep/Hep^ (*n* = 6) independent observations. **h** GTT AUC measured from (**g**). **i** Body weight measured from male *Mc3r*^+/+^ (*n* = 8), *Mc3r*^TB/TB^ (*n* = 9) and *Mc3r*^Hep/Hep^ (*n* = 7) independent observations under chow diets. **j** Body weight measured from female *Mc3r*^+/+^ (*n* = 14), *Mc3r*^TB/TB^ (*n* = 8) and *Mc3r*^Hep/Hep^ (*n* = 9) independent observations under chow diets. Data are represented as mean ± SEM. Groups were compared by ordinary one-way ANOVA followed by Tukey's HSD test (**a–d**, **f**, **h**), two-way ANOVA followed by Tukey's HSD test (**e**, **g**, **i**, **j**). * *p* < 0.05; ** *p* < 0.01; *** *p* < 0.001. Source data are provided as a Source Data file.

*Mc3r*^TB/TB^, and *Mc3r*^Hep/Hep^ genotypes. Log2 fold change (Log2FC) values for all significantly differentially expressed genes were plotted in a scatter plot. Linear regression fit line comparisons between *Mc3r*^TB/TB^ vs. *Mc3r*^+/+^ and *Mc3r*^Hep/Hep^ vs *Mc3r*^+/+^ (blue line, *r* = 0.189, *p* < 0.0001) suggest that *Mc3r*^TB/TB^ and *Mc3r*^Hep/Hep^ are drastically different (Fig. 6h). Linear regression fit lines for the comparisons between *Mc3r*^TB/TB^ vs. *Mc3r*^+/+^ and *Mc3r*^Hep/Hep^ vs *Mc3r*^TB/TB^ (orange line, *r* = 0.726, *p* < 0.0001) slightly overlapped with the best-fit perfect correlation line, indicate that *Mc3r*^+/+^ and *Mc3r*^Hep/Hep^ are largely similar (Fig. 6i). These data suggest that the hepatic changes in differentially expressed genes caused by *Mc3r* global knockout are largely reversed by the recovery of hepatic *Mc3r*.

Next, we performed gene ontology (GO) biological pathway analysis for the comparisons between *Mc3r*^TB/TB^ vs. *Mc3r*^+/+^, *Mc3r*^Hep/Hep^ vs *Mc3r*^+/+^ and *Mc3r*^Hep/Hep^ vs *Mc3r*^TB/TB^. A heatmap of negative adjusted log10 *P* values shows that the *Mc3r*^Hep/Hep^ *and Mc3r*^+/+^ groups had greater similarity (Fig. 6j). In contrast, *Mc3r*^TB/TB^ shows much lower *p*-values that are significantly different compared to *Mc3r*^+/+^ and *Mc3r*^Hep/Hep^, particularly in pathways related to lipid droplet autophagy and lipid metabolism (Fig. 6j). These findings reinforce the notion that liver autophagy is a predominant mechanism influencing fat accumulation and lipid partitioning in MC3R deficiency.

In line with a hepatic role for MC3R and the role of autophagy in lipid metabolism[3,8], *Mc3r*^Hep/Hep^ mice indeed had a diminution of the augmented hepatic LC3II signals to the level seen in control livers (Fig. 6k, l). The liver ratio of LC3II to LC3I was also normalized in *Mc3r*^Hep/Hep^ mice (Fig. 6m), supporting the restored hepatic LC3 development.

## Hepatic *MC3R* activation and circulating plasma neuropeptides in autophagy regulation

Gamma-melanocortin stimulating hormone (γ-MSH), derived from proopiomelanocortin (POMC), has specific affinity to MC3R over MC4R[43]. Therefore, we examined if the circulating γ-MSH pool is altered in mice with deficiency for *Mc3r*. Using an enzyme-linked immunosorbent assay (ELISA) specific to gamma 2 MSH, we found that plasma γ-MSH significantly increased at nighttime (ZT 15) compared to daytime (ZT 3) in control mice (Supplementary Fig. 5a). *Mc3r*^TB/TB^ mice exhibited impaired plasma γ-MSH regulation during the circadian cycle, although fasting could elevate plasma γ-MSH at ZT15 in wildtype mice (Supplementary Fig. 5a). These data suggested altered circadian control of circulating γ-MSH signaling in *Mc3r*-insufficient mice. Further, the primary hepatocytes with defective MC3R pathway signaling failed to produce cAMP, compared to the control cells, in response to the MC3R-specific (versus MC4R)[43] agonist, [D-Trp^8^]-γ-MSH (Supplementary Fig. 5b). Thus, our data suggest that *Mc3r*^TB/TB^ mice have defective MC3R signaling in liver cells, possibly due to loss of both the receptor (Fig. 1b) and ligand functions (Supplementary Fig. 5a).

To examine how activation of the MC3R pathway affects hepatic autophagy, we assessed AP organization in primary hepatocytes that carry transgenic GFP-LC3 with the control or deficient *Mc3r* after stimulation with [D-Trp^8^]-γ-MSH. In control cells, [D-Trp^8^]-γ-MSH significantly increased cytosolic GFP-LC3 structures to a similar extent as seen in the cells exposed to rapamycin, a well-known pharmacological

activator of liver autophagy[44,45] (Fig. 7a). Using live cell imaging, we found dispersed cytoplasmic GFP-LC3 molecules rapidly develop into multiple vesicles (or tubular structures) at 30 min of [D-Trp^8^]-γ-MSH treatment (Fig. 7b). These GFP-LC3 vesicles further expanded, eventually fusing to lysotracker red containing subcellular compartments (i.e., lysosomes). Soon after lysosomal fusion, LC3 fluorescent signals became dimmer, possibly due to degradation, recycling, and dispersal of LC3 molecules (Fig. 7b, See Inlay). Thus, MC3R activation by γ-MSH serves as a signal for reorganizing hepatic LC3 in autophagosomes. To confirm the observed results, we have included representative data from time-lapse live cell imaging of primary hepatocytes treated with the MC3R agonist, gamma MSH for both *Mc3r*-deficient and wild-type mice (Supplementary Fig. 6), again showing greater autophagosome development after hormone stimulation in control mice but not in *Mc3r*^TB/TB^.

Similar to our imaging data, cultured hepatocytes carrying intact *Mc3r* increased LC3II upon [D-Trp^8^]-γ-MSH exposure in a time-dependent manner (Fig. 7c). We also confirmed these findings using a synthetic specific ligand for MC3R (NDP-MSH compound-42, KAF4098-32 (sequence Ac-Val-Gln-(pI)DPhe-DTic-NH2))[46] thus indicating specific involvement of hepatic MC3R signaling in controlling LC3II production in primary hepatocytes (Fig. 7d–g). Importantly, no such increase of LC3II was evident in either *Mc3r*^TB/TB^ (Fig. 7c) or *MC3R*^hDM/hDM^ hepatocytes (Supplementary Fig. 5c), indicating perturbed hepatic autophagy in the absence of intact *Mc3r*.

Similarly, imaging of *MC3R*^hDM/hDM^ hepatocytes revealed perturbed GFP-LC3 vesicle development and interaction with lysosomes in response to MC3R agonists (Supplementary Fig. 5d). To further test if in vivo activation of MC3R can modulate hepatic autophagy, liver LC3II was examined in vivo in *Mc3r*^+/+^ mice (fed state) after intraperitoneal administration of [D-Trp^8^]-γ-MSH (Fig. 7h, i and Supplementary Figs. 5e and 7f). Compared to saline injection, γ-MSH significantly increased liver LC3II (Fig. 7i) in mice with intact MC3R. Therefore, both in vitro and in vivo, activation of MC3R appeared to modulate mouse liver autophagy.

## TFEB signaling affected by MC3R pathway in hepatic autophagy regulation

Several signaling mechanisms that are key to liver autophagy regulation have been proposed. Inhibition of mechanistic target of rapamycin kinase complex I (MTORC1) or activation of AMP-activated protein kinase (AMPK) can trigger liver autophagy[47,48]. In addition, for more sustainable autophagy, the mechanism may involve the activation of TFEB, a master regulator of autophagy and lysosome biogenesis, for which nuclear-cytoplasmic shuttling is required for the upregulation of autophagy- and lysosomal-related genes[49,50].

To explore potential signaling mechanisms that interplay with MC3R for liver autophagy regulation, we investigated MTORC1 and AMPK signaling as well as nuclear-cytoplasmic TFEB signaling. Using primary hepatocytes from control and *Mc3r*^TB/TB^ mice, we found that under serum starvation, 4E-BP1 phosphorylation at Thr37/46 (indicative of MTORC1 activation) was clearly inhibited in both control and *Mc3r* defective hepatocytes (Supplementary Fig. 7a). This suggests a similar degree of autophagy initiation by MTORC1 inhibition occurs in

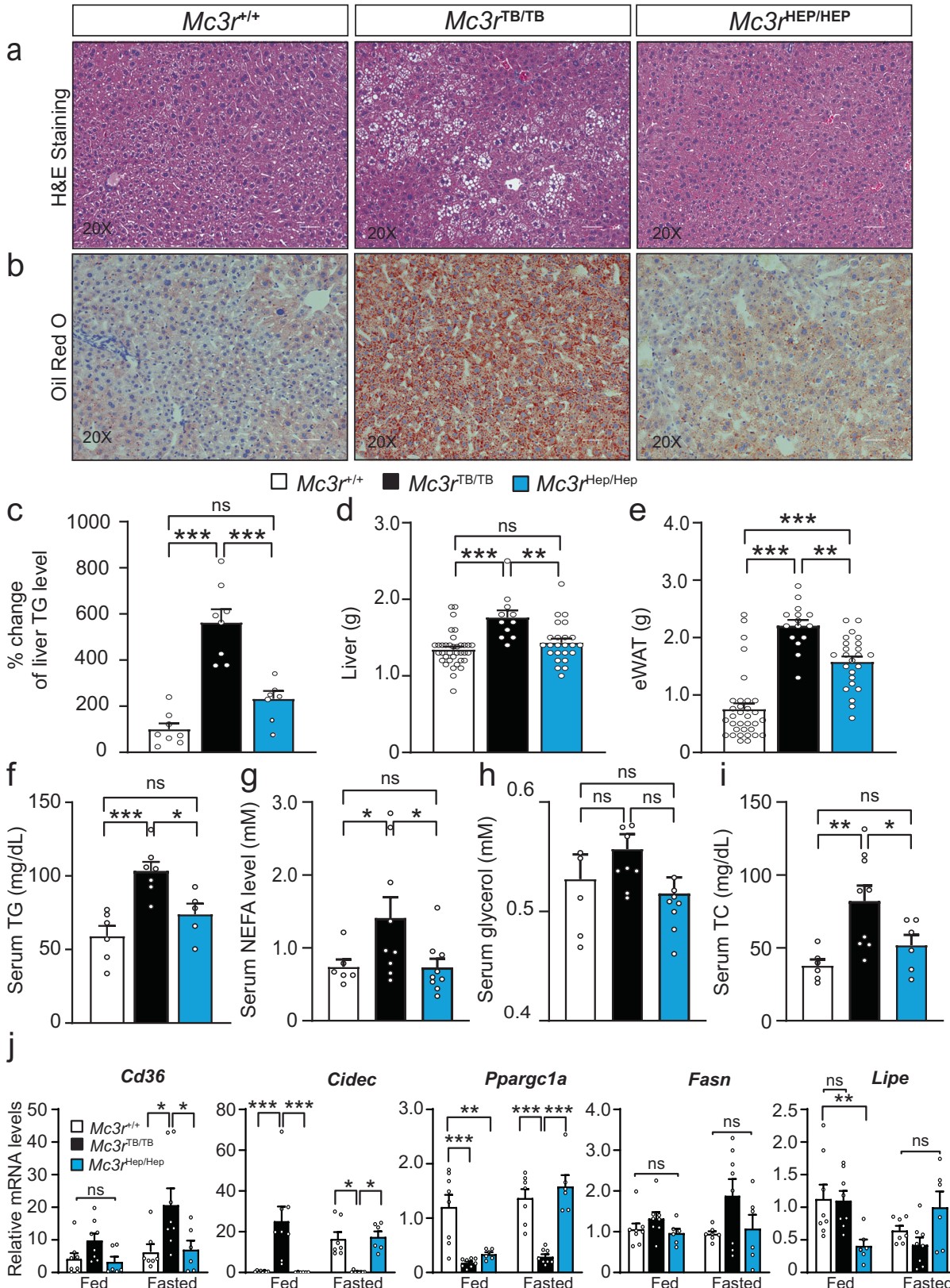

both groups. Serum starvation also did not alter AMPKα phosphorylation (indicative of AMPK activation) in the MC3R defective hepatocytes (Supplementary Fig. 7a), indicating unchanged liver AMPK signaling. However, unlike stimulation of nuclear TFEB localization in the control cells or $Mc3r^{Hep/Hep}$ cells, $Mc3r^{TB/TB}$ hepatocytes failed to trigger TFEB's nuclear delivery during serum starvation (Fig. 7j). Thus,

TFEB shuttling to the nucleus could be triggered by [D-Trp[8]]-γ-MSH in hepatocytes with intact $Mc3r$, but not in the cells from $Mc3r^{TB/TB}$ mice (Fig. 7j), concurrently there was no change in TFEB protein expression in whole cell protein lysates (Supplementary Fig. 2b). These results suggest a potential interplay between MC3R pathway and TFEB signaling for hepatic autophagy.

**Fig. 4 | Hepatic *Mc3r* reactivation reversed fat accumulation and fatty acid metabolism in *Mc3r* insufficient mice. a** H&E staining (**b**) Oil Red O staining for liver tissue from *Mc3r*[+/+], *Mc3r*[TB/TB] and *Mc3r*[Hep/Hep]. **c** Fold change of liver TG levels from *Mc3r*[+/+] ($n = 8$), *Mc3r*[TB/TB] ($n = 8$) and *Mc3r*[Hep/Hep] ($n = 7$) independent observations. **d** Liver weight (**g**) from *Mc3r*[+/+] ($n = 36$), *Mc3r*[TB/TB] ($n = 11$) and *Mc3r*[Hep/Hep] ($n = 24$) independent observations (**e**) eWAT weight (**g**) from *Mc3r*[+/+] ($n = 33$), *Mc3r*[TB/TB] ($n = 15$) and *Mc3r*[Hep/Hep] ($n = 24$) independent observations (**f**) Serum TG measured from *Mc3r*[+/+] ($n = 6$), *Mc3r*[TB/TB] ($n = 7$) and *Mc3r*[Hep/Hep] ($n = 5$) independent observations. **g** Serum non-esterified fatty acids (NEFA) from *Mc3r*[+/+] ($n = 6$), *Mc3r*[TB/TB] ($n = 9$) and *Mc3r*[Hep/Hep] ($n = 9$) independent observations. **h** Serum glycerol from *Mc3r*[+/+] ($n = 6$), *Mc3r*[TB/TB] ($n = 9$) and *Mc3r*[Hep/Hep] ($n = 9$) independent observations. **i** Serum TC from *Mc3r*[+/+] ($n = 6$), *Mc3r*[TB/TB] ($n = 9$) and *Mc3r*[Hep/Hep] ($n = 6$) independent observations. **j** qPCR mRNA measurements for fatty acid metabolism from *Mc3r*[+/+] ($n = 8$ *fed*) *and* ($n = 7$ *fasted*) independent observations, *Mc3r*[TB/TB] ($n = 8$), and *Mc3r*[Hep/Hep] ($n = 6$) independent observations from both fed-fasted conditions. Data are represented as mean ± SEM. Groups were compared by one-way ANOVA followed by Tukey's HSD test (**c**–**f**), one-way ANOVA followed by Fisher's LSD test (**g**–**i**), and two-way ANOVA followed by Tukey's HSD test (**j**). * $p < 0.05$; ** $p < 0.01$; *** $p < 0.001$, Scale bar, 50 μm (**a**, **b**). Source data are provided as a Source Data file.

Finally, to reveal TFEB activity in the liver, we monitored TFEB-activated downstream genes using primary hepatocytes from control and *Mc3r*[TB/TB] mice. Upon treatment with [D-Trp[§]]-γ-MSH, the transcription of autophagy-related genes and genes for lysosomal homeostasis were significantly upregulated in the hepatocytes with intact *Mc3r* but were dysregulated in cells with insufficient MC3R activity (Fig. 7k). To further investigate precise candidate mechanisms underlying the hepatic *Mc3r* reactivation model, we performed DESeq2 analysis on bulk RNA-sequencing data from liver tissue. Notably, TFEB-driven autophagy-related genes, including *Mcoln1* (mucolipin 1), *Neu1* (neuraminidases), and *Sqstm1* (p62 protein), showed increased expression in *Mc3r*[Hep/Hep] mice compared to *Mc3r*[TB/TB] mice (Fig. 7l). Interestingly, lysosomal-associated membrane proteins, *Lamp1* and *Lamp2* showed expression was elevated in *Mc3r*[TB/TB] mice compared to both *Mc3r*[+/+] and *Mc3r*[Hep/Hep] groups (Fig. 7l). Therefore, our data suggest MC3R-dependent transcriptional activation of TFEB signaling and downstream events in the liver.

## Discussion

We provide evidence identifying a previously unknown regulator of hepatocellular autophagy. We show in two mouse models of MC3R insufficiency that autophagic flux is disrupted in a unique manner; we also demonstrate that, when administered in concentrations similar to those that circulate endogenously and that can initiate MC3R-dependent signaling, γ-MSH can activate the hepatic TFEB signaling program. MC3R activation by γ-MSH drives TFEB signaling, leading to the activation of genes related to autophagy or lysosomal homeostasis. A specific MC3R agonist, NDP-42, also activated hepatic LC3II in *Mc3r*[+/+] hepatocytes but did not further activate LC3II in *Mc3r*[TB/TB], which showed blunted autophagosome-lysosome docking and LD-clearance, leading to defective lipid autophagy, thus confirming the role of MC3R pathway in hepatocytes and not through the activation of other receptors. Liver RNA-seq analysis revealed that differentially expressed genes (DEGs) were significantly altered in global knockout mice, while the DEG profile of hepatic reactivation mice was closer to that of the control group. Gene ontology (GO) pathway analysis suggests that the affected pathways are related to lipid droplet autophagy, macroautophagy, and lipid metabolism, indicating that *Mc3r* deficiency leads to distinct changes compared to either *Mc3r* reactivation or wild-type mice. In line with this notion, recent studies using thermal proteome profiling revealed that the top reactome enriched after MC3R activation is that of autophagy proteins[51]. Importantly, liver-specific MC3R reactivation (*Mc3r*[Hep/Hep] mouse) was sufficient to reestablish the dysregulated TFEB signaling and improve (though not completely restore) systemic lipid metabolism of *Mc3r* null mice. Thus, the current study elucidates a previously unknown function of hepatic MC3R in regulating body weight in mice. These data do not, however, indicate that the MC3R is involved in the stimulation of hepatic autophagy by fasting or starvation. Indeed, it appears that peripheral ACTH concentrations (and therefore likely peripheral gamma-MSH concentrations) are reduced in mice during prolonged food restriction or fasting[52,53].

*Mc3r* expression in the lateral hypothalamic area and ventromedial hypothalamic area play crucial roles in regulating feeding, rheostasis, body weight maintenance, and metabolism[23,25,54,55]. Our data confirmed relative hypophagia in transcriptionally blocked global *Mc3r* deficient mice studied in metabolic chambers (using real-time monitoring feeder hangers)[23]. We found that the recovery of liver-specific *Mc3r* did not reverse lowered energy intake (as indicated in Fig. 5a–c), and thus, hepatic re-expression of *Mc3r* did not restore feeding behavior, supporting the previous finding that Mc3r deficiency is related to appetite regulation by hypothalamic MC3R neurons[23]. However, total energy expenditure, oxygen consumption and RER were reduced in global *Mc3r* deficient mice even after adjusting for body weight and total lean plus fat mass. Total energy expenditure measured using indirect calorimetry was lower in *Mc3r*[TB/TB] in the mid-dark period and even after adjustment for body weight[23]. Hepatic *Mc3r* reactivation improved total energy expenditure, oxygen consumption, and RER. These differences in total energy expenditure, oxygen consumption, and RER, as observed here, may contribute to the less obese body composition of *Mc3r*[Hep/Hep]. RER is an indicator of oxidizing fatty acids and is significantly lower in *Mc3r*[TB/TB]. The increased RER in *Mc3r*[Hep/Hep] suggests a shift in substance preference. Hepatic reactivation of *Mc3r* restored lipid recycling and shifted metabolism towards using carbohydrates for fuel. Improvement in energy expenditure-related parameters suggests a potential mechanism to improve body weight, fat mass, and systemic adiposity.

In a similar line, we assessed cellular respiration in primary hepatocytes. *Mc3r* deficient mice exhibited reduced oxygen consumption, suggesting a reduced metabolic demand and a decrease in their maximal electron transport capacity, as these cells are not fully utilizing their mitochondrial capacity under normal conditions. Notably, in red oxidative muscle isolated from gastrocnemius, *Mc3rKO* mice were previously shown to have a decrease in fatty acid oxidation and citrate synthase activity, consistent with reduced mitochondrial content[14].

Primary hepatocytes isolated from mice exposed to a 60% high-fat diet (HFD) for 12 weeks (who generally develop diabetes) have been reported to have reduced basal respiration as well as reduced ATP-linked respiration (oxidative phosphorylation), and their response to the uncoupler FCCP was also decreased when mitochondrial energy metabolism assessed via extracellular flux analysis[56]. However, mitochondria isolated from mice with liver steatosis from a 35% high-fat and high-sugar diet for 5 months (who mostly show insulin resistance without diabetes) have normal levels of mitochondrial respiratory chain complex I–V proteins and do not demonstrate adversely impacted mitochondrial respiration[57]. These data suggest that hepatic mitochondria may initially activate compensatory mechanisms to counteract liver damage associated with severe hepatic steatosis. However, persistent fat accumulation can eventually surpass this adaptive capacity, leading to impaired mitochondrial function due to an overload of free fatty acids[58]. Indeed, primary human hepatocytes exposed in vitro to FFAs showed suppressed maximal respiration and maximum fatty acids beta-oxidation, further indicating compromised mitochondrial function[59]. Consequently, reductions in liver mitochondrial State 3 respiration can be found[60], particularly in mice with high histological grade MAFLD[61]. However, there are also rodent models, e.g., the Otsuka Long-Evans Tokushima fatty rat[62] and mice

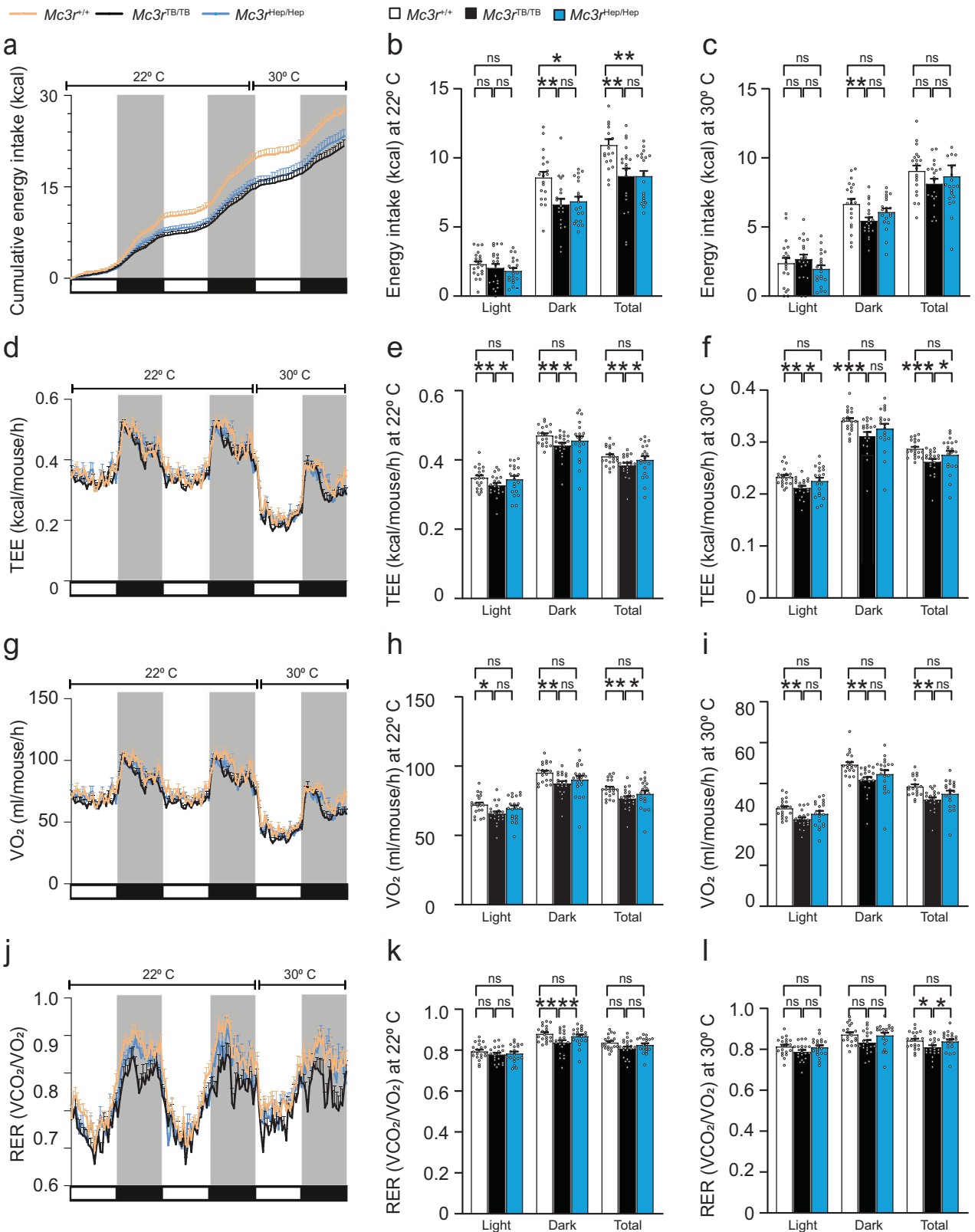

with heterozygous inactivation of the mitochondrial trifunctional protein[63], where mitochondrial dysfunction is present before insulin resistance and steatosis develops, and is believed to contribute etiologically to the development of MAFLD.

Dietary fats absorbed by the gut are transported to the liver for proper processing, packaging into lipoproteins, and circulation. In the liver, autophagy is the primary mechanism for the catabolism of

lipoproteins, although lipolysis also occurs, though to a lesser extent. Reactivation of *Mc3r* leads to improved lipid droplet autophagy; here, we showed how *Mc3r* insufficiency affected cellular and molecular mechanisms of autophagy. Due to defective *Mc3r*-mediated lipid droplet autophagy, hepatocytes do not efficiently metabolize triglycerides. By restoring defective autophagy, we observed reductions in liver triglyceride content and liver weight and partial

**Fig. 5 | Assessment of energy intake and energy expenditure at ambient and thermoneutral temperatures by mice indirect calorimetry. a** Cumulative energy intake (kcal) monitored for light and dark phase at ambient 22 °C and thermoneutral temperature 30 °C. **b** Daily energy intake (kcal) for light, dark, and total (24 h) at 22 °C from $Mc3r^{+/+}$, $Mc3r^{TB/TB}$ and $Mc3r^{Hep/Hep}$ ($n = 20$) independent observations. **c** Daily energy intake (kcal) for light, dark, and total (24 h) at 30 °C from $Mc3r^{+/+}$, $Mc3r^{TB/TB}$, and $Mc3r^{Hep/Hep}$ ($n = 20$) independent observations. **d** Total energy expenditure (kcal/mouse/h) for light and dark phase **e** TEE (kcal/mouse/h) for light, dark, and total (24 h) at 22 °C from $Mc3r^{+/+}$, $Mc3r^{TB/TB}$, and $Mc3r^{Hep/Hep}$ ($n = 20$) independent observations. **f** TEE (kcal/mouse/h) for light, dark, and total (24 h) at 30 °C from $Mc3r^{+/+}$, $Mc3r^{TB/TB}$, and $Mc3r^{Hep/Hep}$ ($n = 20$) independent observations. **g** Oxygen consumption (VO₂) (ml/mouse/h) for light and dark phase **h** VO₂ (ml/mouse/h) for light, dark, and total (24 h) at 22 °C from $Mc3r^{+/+}$, $Mc3r^{TB/TB}$, and $Mc3r^{Hep/Hep}$ ($n = 20$)

independent observations. **i** VO₂ (ml/mouse/h) for light, dark, and total (24 h) at 30 °C from $Mc3r^{+/+}$, $Mc3r^{TB/TB}$, and $Mc3r^{Hep/Hep}$ ($n = 20$) independent observations. **j** Respiratory exchange ratio (RER) (VCO₂/VO₂) for light and dark phase (**k**) RER (VCO₂/VO₂) for light, dark, and total (24 h) at 22 °C from $Mc3r^{+/+}$, $Mc3r^{TB/TB}$, and $Mc3r^{Hep/Hep}$ ($n = 20$) independent observations. **l** RER (VCO₂/VO₂) for light, dark, and total (24 h) at 30 °C from $Mc3r^{+/+}$, $Mc3r^{TB/TB}$, and $Mc3r^{Hep/Hep}$ ($n = 20$) independent observations. In (**a**, **d**, **g**, and **j**), data from the first day of adaptation are not shown. Chow-fed, $n = 20$ mice/group (10 males and 10 females) mice 3–4 months of age. Data are represented as mean ± SEM. Groups were compared using Friedman's test by Dunn's paired analysis (**b**, **c**, **e**, **f**, **h**, **i**, **k**, **l**) and post hoc test for adjusted estimated marginal means in Supplementary Table 1. * $p < 0.05$; ** $p < 0.01$; *** $p < 0.001$. Source data are provided as a Source Data file.

improvements in eWAT weight. Our results demonstrated the restoration of normal circulating non-esterified fatty acid concentrations, total cholesterol, and triglycerides in $Mc3r^{Hep/Hep}$ mice. These suggest that $Mc3r$ reactivation largely restores lipid metabolism in hepatocytes and leads to reduced excess circulating fats that would be stored by extrahepatic tissues (e.g., eWAT). The partial recovery of body weight, fat mass, and total energy expenditure at the systemic level underscores how hepatic $Mc3r$ reactivation can modulate a systemic obesity phenotype.

The restoration of hepatic $Mc3r$ resulted in improvements in adiposity, metabolic dysfunction-associated steatotic liver disease, and insulin sensitivity. Our observations from $Mc3r^{TB/TB}$ and $MC3R^{hDM/hDM}$ livers lead us to hypothesize that the MC3R pathway functions specifically in determining lysosomal-AP interaction and degradation processes, based on the: (1) prevalent AP accumulation with lumenal compartments containing undegraded organellar membranes (Fig. 2a–c and h, i), (2) impaired AP flux and p62 clearance upon serum starvation (Fig. 2f, g and Supplementary Fig. 1c), and (3) increased basal LC3II in both hepatic tissues and isolated hepatocytes (Fig. 2d, e). In addition, phopho-4E-BP1 degradation confirms the LC3II activation and change in MTORC1 might be possible upstream autophagy mediators (Supplementary Fig. 7a). Moreover, the findings of MC3R-dependent nuclear TFEB allocation (Fig. 7j) with subsequent autophagy and lysosomal gene activations (Fig. 7k) further support MC3R's specific role in the regulation of autophagy.

Given hepatic TFEB's central importance in lysosomal biogenesis[49,64–66], the pathogenic outcome of defective MC3R signaling in the liver might have been a result of impaired lysosome development. However, our fluorescent imaging data indicates no apparent defect in lysosome structures in MC3R defective livers (Fig. 2h). In addition, the data obtained from monitoring cathepsin D (a lysosomal aspartyl protease) maturation shows intact lysosomal trafficking and development without MC3R activity in hepatocytes (Supplementary Fig. 7b). Therefore, MC3R likely plays a role in fine-tuning lysosomal activity, rather than driving overall lysosomal biogenesis. Since loss of $Mc3r$ leads to TFEB accumulation in the nucleus while reducing γ-MSH-induced TFEB nuclear localization (Fig. 7j), data shown here that MC3R signaling possibly modulates hepatic lysosome activity through the control of TFEB nuclear import and export activities. We also showed that expression of TFEB downstream target genes for autophagy-related and lysosomal-associated membrane proteins is restored in hepatic $Mc3r$ re-expression. Further research is needed to determine whether TFEB is the sole regulator responsible for mediating autophagy regulation in MC3R deficiency. In the future, more comprehensive research on MC3R-autophagy-TFEB regulation is warranted.

Finally, our findings of $Mc3r$'s peripheral role in hepatic autophagy opens opportunities in both translational and basic research. Despite the beneficial effect of liver autophagy in preventing human liver metabolic diseases[39,67], enhancing hepatocellular autophagy by non-physiological agonists (e.g., rapamycin) or gene manipulation has

limited merit due to side effects or limited clinical applicability. Thus, the demand for developing tools for liver autophagy regulation with physiologically approachable and effective means is high, and this makes peripheral modulation of the MC3R a conceivable strategy. In order to delve deeper into the impact of $MC3R$ on hepatic lipid droplet autophagy, it will be necessary to utilize a mouse model with a tissue-specific knockout. Furthermore, the dual roles of MC3R in the neuronal and peripheral tissues may also provide a model system to study biological impacts and mechanisms of the brain-peripheral tissue communication in metabolic regulation.

Taken together, our study provides insight into the role of peripheral MC3R in regulating body weight and adiposity in mice. Given the similar obesogenic mechanism seen in our humanized $MC3R$ mutant mouse model, insufficient peripheral activation of MC3R appears to likely play a part in explaining obesity in humans with MC3R deficiency.

## Methods

### Materials and reagents
Information on reagents, antibodies, primers are provided in Supplementary Table 2. Datasets are publicly available[68].

### Ethics statement
This research complies with all relevant ethical regulations. All animal studies were reviewed and approved for the accepted standards of humane animal care under protocols approved by the NICHD Animal Care and Use Committee.

### Animals
Studies were performed in males and female C56BL/6 background mice with the following genotypes: $Mc3r^{TB/TB}$, $MC3R^{hDM/hDM}$, $MC3R^{hWT/hWT}$, $MC3R^{Hep/Hep}$, $Mc4r^{+/-}$ and wild-type (BL/6). GFP-LC3 mice that were cross bred with $MC3R^{TB/TB}$, $MC3R^{hDM/hDM}$, $MC3R^{hWT/hWT}$, and WT mice were a gift from Dr. Noboru Mizushima (RIKEN Bio, Japan), who first developed this model in 2008, and the breeding pair was received from Dr. Sergey Leikin of the NICHD, an investigator who currently has a colony of GFP-LC3 mice. Albumin-Cre mice were purchased from Jackson Laboratory (B6.Cg-Speer6-ps1Tg(Alb-cre)21Mgn/J, stock#:003574) and crossed with $MC3R^{TB/TB}$ to generate $MC3R^{Hep/Hep}$ (Supplementary Table 3). Mice were housed and maintained on a 14-h light, 10-h dark cycle and studied at 21–25 °C. We monitored body weight from 4 to 24 weeks of age. Mice were fed a regular chow NIH-07 diet containing metabolizable 3.05 kcal/g (Lab Diet, Arden Hills, MN), and tissue samples were collected from mice that were either at 12 or 22 weeks of age. Mice were randomly selected for different groups or different treatments.

### Mouse body composition
Body composition was determined using a PIXImus dual-energy X-ray absorptiometer (DEXA, Lunar, Madison, WI). Mice were either euthanized with CO₂ or anesthetized with ketamine and xylazine (100/10 mg/kg body weight) by intraperitoneal injection.

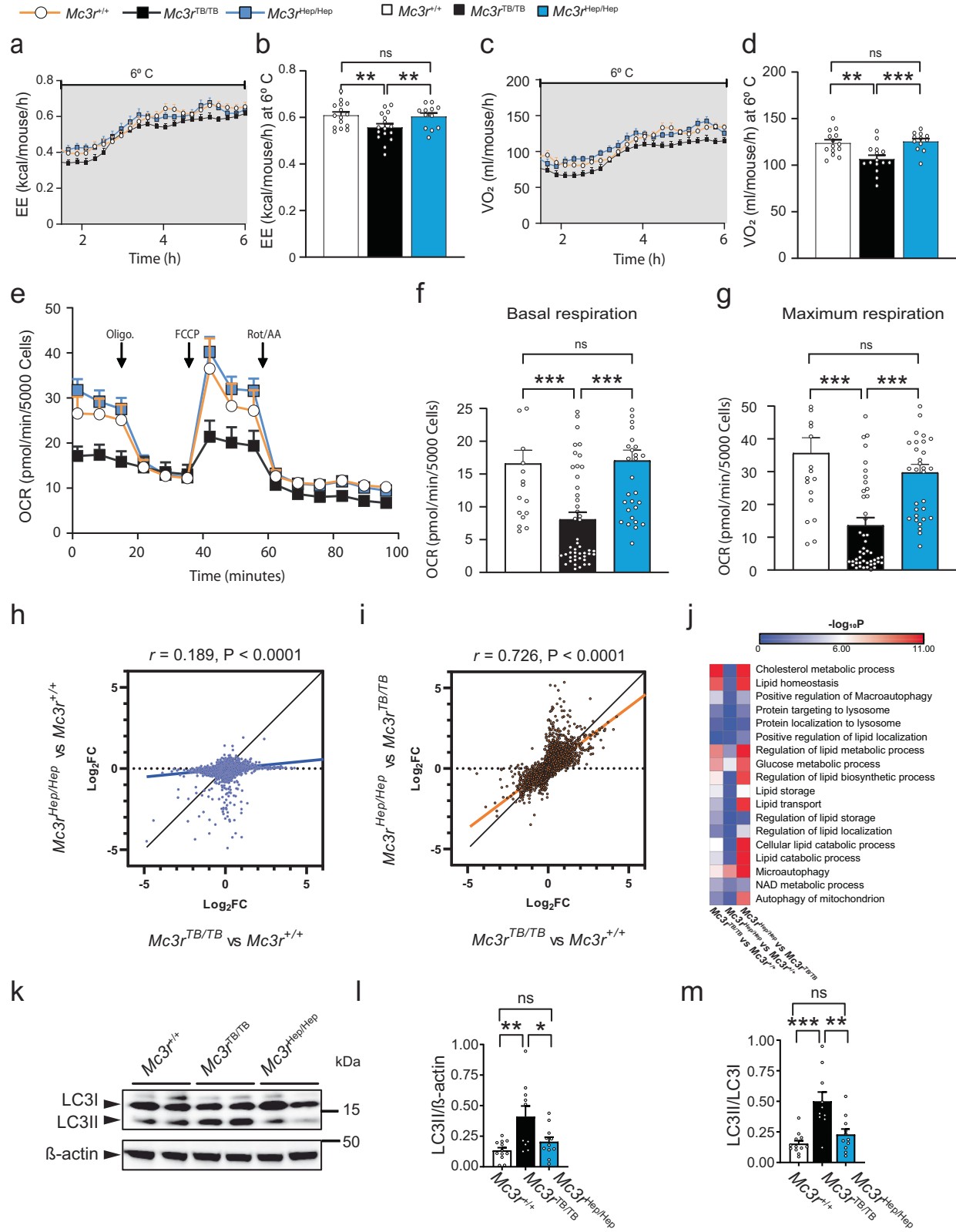

## Insulin tolerance test

Mice were fasted for 5 h before an i.p. administration of 0.75 U/kg body weight of insulin. The load glucose of each mouse was measured using a human glucometer with blood drops from the tail vein after tail snipping. Basal glucose was determined at the 0 time point, and blood glucose was measured 30, 60, 90, and 120 min after insulin was injected.

## Glucose tolerance test

Mice were fasted for 16 h (overnight) prior to glucose tolerance testing. Baseline blood glucose was measured from a tail snip blood droplet collection using a Contour NEXT EZ meter with Contour NEXT Test strips (Ascensia Diabetes, Parsippany, NJ). Mice were then given 2.5 g/kg body weight by intraperitoneal glucose

**Fig. 6 | Hepatic *Mc3r* reactivation restores adaptive energy expenditure, cellular respiration, and lipid droplet autophagy. a** Energy expenditure (kcal/mouse/h) at 6 °C from *Mc3r*$^{+/+}$ ($n = 15$), *Mc3r*$^{TB/TB}$ ($n = 16$), and *Mc3r*$^{Hep/Hep}$ ($n = 12$) independent observations. **b** EE (kcal/mouse/h) at 6 °C from *Mc3r*$^{+/+}$ ($n = 15$), *Mc3r*$^{TB/TB}$ ($n = 16$), and *Mc3r*$^{Hep/Hep}$ ($n = 12$) independent observations. **c** Oxygen consumption (VO$_2$) (ml/mouse/h) at 6 °C from *Mc3r*$^{+/+}$ ($n = 15$), *Mc3r*$^{TB/TB}$ ($n = 16$), and *Mc3r*$^{Hep/Hep}$ ($n = 12$) independent observations. **d** VO$_2$ (ml/mouse/h) at 6 °C from *Mc3r*$^{+/+}$ ($n = 15$), *Mc3r*$^{TB/TB}$ ($n = 16$), and *Mc3r*$^{Hep/Hep}$ ($n = 12$) independent observations. **e** Oxygen consumption rate (OCR) from cultured primary hepatocytes from *Mc3r*$^{+/+}$, *Mc3r*$^{TB/TB}$ and *Mc3r*$^{Hep/Hep}$ from 3 independent experiments. **f** Basal respiration OCR (pmol/min/5000 cells) from 3 independent experiments. **g** Maximal respiration OCR (pmol/min/5000 cells) from 3 independent experiments. **h** Bulk RNA-Sequencing analysis of mouse liver samples from *Mc3r*$^{+/+}$, *Mc3r*$^{TB/TB}$, and *Mc3r*$^{Hep/Hep}$ genotypes, all on a normal chow diet ($n = 4$) independent observations. Pairwise scatter plot illustrates the Log$_2$ fold change (Log$_2$FC) values for all significantly differentially expressed genes across three genotype comparisons. The Log$_2$FC for *Mc3r*$^{TB/TB}$ vs *Mc3r*$^{+/+}$ is plotted against *Mc3r*$^{Hep/Hep}$ vs *Mc3r*$^{+/+}$ (blue-filled circles), and linear regression fit lines are shown for the comparisons between *Mc3r*$^{TB/TB}$ vs. *Mc3r*$^{+/+}$ and *Mc3r*$^{Hep/Hep}$ vs *Mc3r*$^{+/+}$ (blue line). **i** The Log$_2$FC for *Mc3r*$^{TB/TB}$ vs *Mc3r*$^{+/+}$ is plotted against *Mc3r*$^{Hep/Hep}$ vs *Mc3r*$^{TB/TB}$ (orange-filled circles). Linear regression fit lines are shown for the comparisons between *Mc3r*$^{TB/TB}$ vs. *Mc3r*$^{+/+}$ and *Mc3r*$^{Hep/Hep}$ vs *Mc3r*$^{TB/TB}$ (orange line), with the black line shows best fit perfect correlation lines included for visualization. **j** Heatmap depicting significant gene ontology (GO) biological pathways related to lipid droplet autophagy, lipid metabolism, and energy homeostasis. The negative log$_{10}$ $P$ values for these pathways were compared across the *Mc3r*$^{TB/TB}$ vs. *Mc3r*$^{+/+}$, *Mc3r*$^{Hep/Hep}$ vs *Mc3r*$^{+/+}$ and *Mc3r*$^{Hep/Hep}$ vs *Mc3r*$^{TB/TB}$ comparisons (adjusted *p-value* ≤ 0.05). **k** Western blot images of liver lysates from *Mc3r*$^{+/+}$, *Mc3r*$^{TB/TB}$ and *Mc3r*$^{Hep/Hep}$. **l** Quantification of LC3II normalized by β-actin in (**g**) from *Mc3r*$^{+/+}$ ($n = 12$), *Mc3r*$^{TB/TB}$ ($n = 10$), and *Mc3r*$^{Hep/Hep}$ ($n = 11$) independent observations. **m** Quantification of LC3II normalized by LC3I in (**g**) from *Mc3r*$^{+/+}$ ($n = 12$), *Mc3r*$^{TB/TB}$ ($n = 10$), and *Mc3r*$^{Hep/Hep}$ ($n = 11$) independent observations. Data are represented as mean ± SEM. Groups were compared by one-way ANOVA followed by Fisher's LSD test (**b, d**), one-way ANOVA followed by Tukey's HSD test (**f, g**), two-tailed simple linear regression (**h, i**). $p < 0.05$; ** $p < 0.01$; *** $p < 0.001$. Source data are provided as a Source Data file.

(Sigma-Aldrich, St. Louis, MO, #G7021) injection. Blood glucose measurements were taken every 30 min for 2 h.

### Food intake and feeding efficiency monitoring
12 week old mice were single-housed for 15 days, with food weight and body weight measured every 3 days. Total food intake was recorded for the 15 day period and then divided by total change in body weight to determine the feeding efficiency. After monitoring was complete, mice were then returned to their original cagemates.

### Energy expenditure, energy intake, and locomotion activity
Total energy expenditure (TEE), energy intake (calculated using metabolizable energy of diet 3.05 kcal/g), oxygen consumption rate (VO$_2$), respiratory exchange ratio (RER), and physical activity (infrared beam breaks as total activity, 1-inch spacing) were measured with an indirect calorimetry system (CLAMS-HC Oxymax v5.52 software, Columbus Instruments) as previously described[69]. Prior to the metabolic measurements, mice were acclimated in the metabolic chambers at room temperature for 2 days, so the data recorded on the second day were excluded from the daily energy expenditure final calculations. On day 3, metabolic parameters were recorded (260 s intervals) continuously at 22 °C for 24 h. On day 4, the chamber temperature was elevated to 30 °C at 6 am and recorded for additional 24 h (the first hour after temperature change was excluded from analysis). During the second week, mice were acclimated to 22 °C in metabolic chambers for 24 h, followed by cold exposure at 6 °C for 6 h, recorded from morning to noon. Mice were housed individually with ad libitum access to food (hanging feeder) and water in Tecniplast 1284 cages with ~95 g of wood chip bedding (7090 Teklad sani-chips, Envigo, Indianapolis, IN) with measured physical activity (infrared beam breaks as total activity, 1 inch spacing) and continual monitoring of total activity in each cage. Calorimetry parameters are: 7.75 L volume, 0.9 L/min flow rate, 0.6 L/min sampling flow, 15 s settle time, 5 s measure time, with each chamber sampled every 260 s. Thus, the physical activity was measured per 260 s interval, giving 14 sampling cycles per 61 min interval. All 12 calorimetry chambers were housed in a single temperature-controlled environmental chamber.

### Liver perfusion and primary hepatocyte culture
Mice were anesthetized at 8 weeks of age for a terminal surgery to collect hepatocytes following hepatic perfusion. The hepatic portal vein and inferior vena cava were exposed by carefully moving the viscera to the right, outside of the abdominal cavity. A 20-gauge catheter was inserted into the inferior vena cava and the perfusion tubing was connected to the needle. Perfusion was initiated at a 4 ml/min flow rate with about 25–30 ml of pre-warmed (37 °C) Liver Perfusion Medium (GIBCO 17701-038). Once successful cannulation was confirmed, the portal vein was cut to allow efflux and the superior vena cava was clamped. The liver became pale in color after 5 to 8 min of perfusion, and the solution was switched to Liver Digest Medium (GIBCO 17703-034), containing type I collagenase for 10 min at a 4 ml/min flow rate. Pressure was applied periodically (5–10 times during the procedure) with a swab to the portal vein for 5-s intervals so that the liver swelled, leading to enhanced hepatic cell dissociation, which in turn increased final yield. After collagenase perfusion, when the liver became porous and spongy in texture, the liver was harvested and placed in a pre-chilled sterile petri dish with Hepatocyte Wash Medium (GIBCO 17704-024) followed by tissue cell culture using Waymouth medium (GIBCO 11200-035) containing 3% FBS, 1% Insulin-Transferrin-Selenium and 1% Penicillin-Streptomycin. Cultured hepatocytes were treated for further assay after 24–48 h.

### Seahorse mitochondria stress test
Primary hepatocytes were imaged and de-gassed using the BioTek Cytation 5 Imaging Reader. Seahorse XF Cell Mito Stress Test (Cat#103015-100) was performed with approximately 5000 cells per well using the Agilent Seahorse XF Pro with our optimized concentrations of 1 µM oligomycin, then 1 µM Carbonyl cyanide-4 (trifluoromethoxy) phenylhydrazone (FCCP), then 0.5 µM rotenone and 0.5 µM antimycin A, and finally 2.5 µM Hoechst. Data analysis was performed using Seahorse Wave Desktop Software, which adjusts for differences in cell number within each well studied.

### Total protein isolation and protein subcellular fractionation
Protein was isolated from liver tissues or cultured hepatocytes using RIPA buffer containing a proteinase inhibitor. Tissues or cells were sonicated and centrifuged at $13,000 \times g$ for 15 min at 4°. Supernatant was collected for total protein and protein concentrations were measured by BCA assay. For subcellular fractionation, primary hepatocytes were harvested after the treatment and washed with 1×PBS with protease-phosphatase inhibitor cocktail on ice. Hepatocytes were lysed in buffer A [10 mM HEPES buffer (Ph 7.8), 0.34 M sucrose, 10% glycerol, 10 mM KCl, 1.5 mM MgCl2, 1 mM PMSF, protease and phosphatase inhibitors, and 0.1% Triton-X100] by repeated pipetting and kept on ice for 7 min to generate the cytoplasmic proteins were separated from nuclei by centrifuging at $2000 \times g$ for 5 min at 4 °C. The supernatant was collected in separate tubes as a cytosolic fraction and the pellet was washed with buffer A and centrifuged at $1500 \times g$ for 5 min. The resultant nuclear pellet was suspended in nuclear lysis buffer B [50 mM Tris-HCl, pH 7.8, 420 mM NaCl, 0.5% IGEPAL, 0.34 M sucrose, and protease and phosphatase

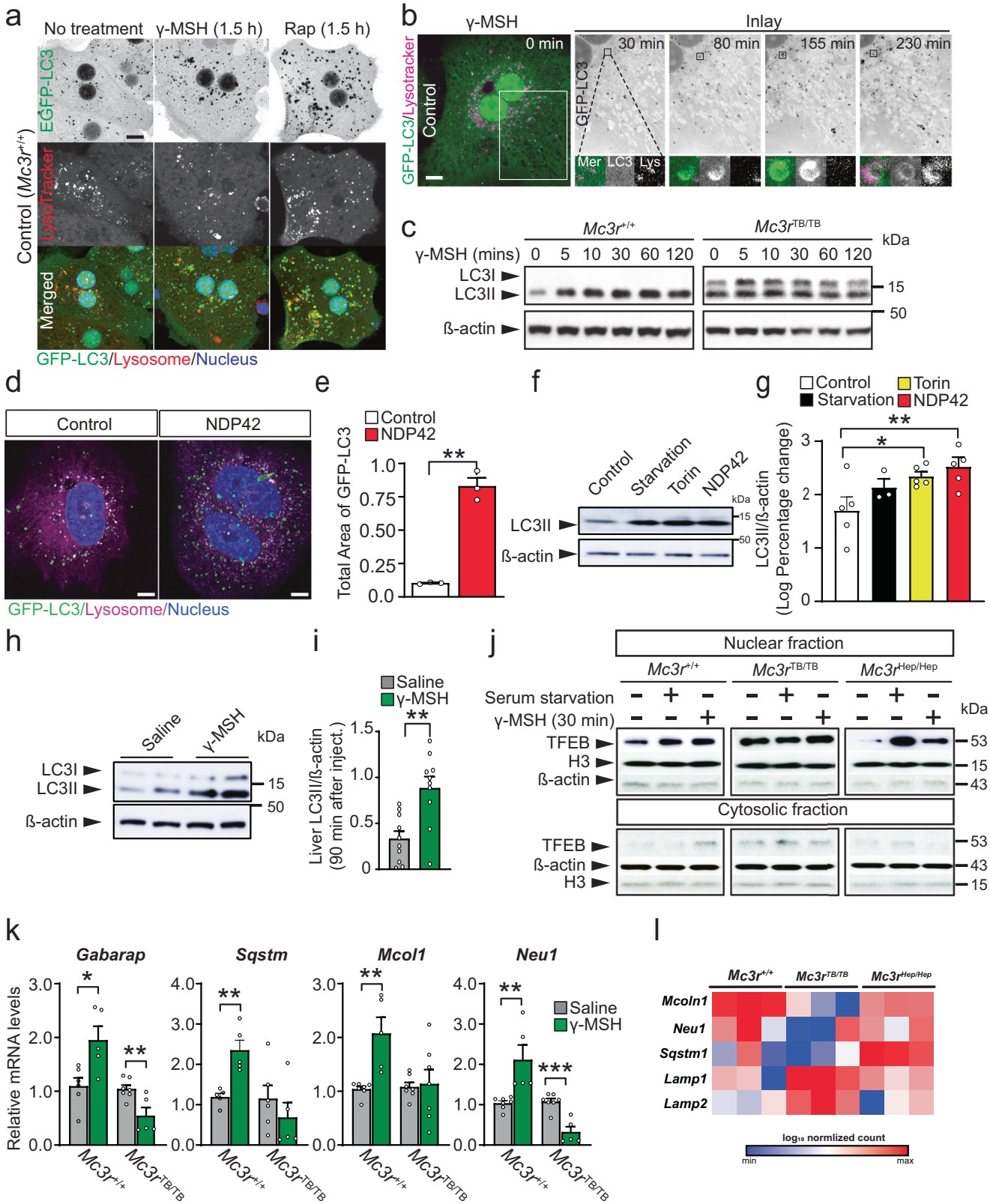

inhibitors]. Nuclei were kept on ice for 30 min and then spun at 16,000 × g for 30 min at 4 °C to separate the nucleoplasm.

## Western blot assay

Protein samples were separated using 4–12% NuPAGE gels (Invitrogen, Carlsbad, CA) and the membranes were blotted with antibodies for rabbit polyclonal anti-LC3I/II, rabbit polyclonal anti-p4E-BP, rabbit polyclonal MTORC1, rabbit polyclonal anti-pAMPK, rabbit polyclonal anti-AMPK, mouse monoclonal anti-β-actin, rabbit polyclonal anti-p62, rabbit polyclonal anti-TFEB, rabbit polyclonal anti-H3, rabbit anti-UCP-1, and rabbit anti-vinculin antibodies (Supplementary Table 2). Primary antibodies were diluted to a 1:1000 concentration,

**Fig. 7 | γ-MSH induces hepatic autophagy and nuclear TFEB activity.**
**a** Representative fluorescent images of primary hepatocytes from *Mc3r*⁺/⁺ transgenic mice carrying GFP-LC3 under 2 nM D-Trp⁸-γ-MSH (γ-MSH) or 10 nM rapamycin (Rap) after staining with Lysotracker Red and DAPI. **b** Time-lapse images of primary hepatocytes from *Mc3r*⁺/⁺ transgenic mice carrying GFP-LC3 under 2 nM γ-MSH after staining with Lysotracker Red. **c** Representative western blot images of primary hepatocytes from *Mc3r*⁺/⁺ and *Mc3r*^TB/TB after 2 nM γ-MSH treatment.
**d** Representative super resolution microscope images of primary hepatocytes from *Mc3r*⁺/⁺ transgenic mice carrying GFP-LC3 under 100 nM MC3R-specific (versus MC4R) agonist (NDP42) for 60 min after staining with Lysotracker Red compared to control. **e** Measurements of total area of GFP-LC3 after 60 min of stimulation (**d**).
**f** Representative western blot images of primary hepatocytes treated with serum starvation (HBSS) for 60 min, 1 μM Torin for 60 min, or 100 nM NDP42 (MC3R-specific agonist versus MC4R) for 30 min. **g** Quantification of LC3II normalized by β-actin in (**f**). Groups were analyzed between Control (*n* = 5), Starvation (*n* = 3), Torin

(*n* = 5), and NDP42 (*n* = 5) independent experiments. **h** Representative western blot images of liver lysates from *Mc3r*⁺/⁺ with intraperitoneal γ-MSH injection.
**i** Quantification of LC3II normalized by β-actin in (**h**) (*n* = 10) independent observations. **j** Representative western blot images of nuclear fractions (top panels) and cytosolic fractions (bottom panels) from primary *Mc3r*⁺/⁺ hepatocytes after 1 h starvation (HBSS) or 2 nM γ-MSH treatment. **k** qPCR measurement from *Mc3r*⁺/⁺ and *Mc3r*^TB/TB hepatocytes after 2 nM γ-MSH (*n* ≥ 5 independent observations).
**l** Heatmap of Log10 normalized counts for TFEB target genes and lysosomal genes in mouse liver RNAseq analysis across *Mc3r*⁺/⁺, *Mc3r*^TB/TB, and *Mc3r*^Hep/Hep genotypes on a normal chow diet (*n* = 3) independent observations. Data are represented as mean ± SEM. Groups were compared by paired two-tailed Student's *t*-test (**e**), unpaired two-tailed Student's *t*-test (**i**, **k**), one-way ANOVA followed by Fisher's LSD test (**g**). *p* < 0.05; ** *p* < 0.01; *** *p* < 0.001, Scale bar, 10 μm (**a**, **b**, **d**). Source data are provided as a Source Data file.

and secondary antibodies were diluted to a 1:2500 concentration. Protein levels were quantified by image density scanning using Image J analysis (NIH, Bethesda, MD). The values were adjusted for β-actin expression.

## Brown adipose tissue (BAT) immunofluorescence procedure
Standard protocols for immunofluorescence of BAT were followed with some adaptations. Mice were housed at either 22 °C or exposed to 6 °C for 6 h. After temperature exposure, mice were perfused using 1× PBS followed by 4% paraformaldehyde (PFA); then, BAT samples were fixed overnight in 4% PFA at 4 °C. Cryoprotection of BAT was achieved by immersion in 10%, 20%, and 30% (w/v) sucrose overnight, followed by embedding in OCT and freezing using Tissue-Tek Cryo 3 Flex Cryostat for sectioning. For UCP-1 expression detection, 10 μm frozen sections were airdried for 30 min at room temperature. Sections were blocked with 5% horse serum and incubated in a humidified chamber with UCP-1 (1:250) and perilipin (1:500) antibodies overnight at 4 °C (Supplementary Table 2). Following washing steps, anti-rabbit Alexa Fluor 488 (Thermo Fisher Scientific #A1100) and mouse Alexa Fluor 647 (Thermo Fisher Scientific # A21236) secondary antibodies (1:500) were incubated for 30 min. Following washing steps, slides were treated with DAPI solution (300 nM in 0.1% TBS-T) for 3 min at room temperature to visualize cell nuclei and mounted with VectoMount express mounting medium (Vector Laboratories, Inc). Imaging was performed using a fluorescence microscope at 20× magnification.

## Tissue Hematoxylin and Eosin staining
At 22 weeks of age, mice were anesthetized with ketamine-xylazine mix and perfused with 1× PBS followed by 4% PFA. After 24 h 4% PFA fixation tissued transfer to 70% ethanol. Paraffin sections of liver and eWAT were prepared on slides at 5–10 μm. Stain with hematoxylin and eosin according to standard protocols (Tissue TEK Prisma Stain K, SKU#6190).

## Tissue Oil Red O staining
Frozen sections of liver tissue were prepared on slides at 5–10 μm. The sections were fixed in cold 10% formalin for 10 min followed by rinsing and drying 3 times. The slides were incubated with propylene glycol for 5 min to avoid carrying water into Oil Red O. Slides were stained in a pre-warmed Oil Red O solution for 10 min in 60 °C, then moved to an 85% propylene glycol solution for 5 min. After washing 3 times, the slides were placed in distilled water and covered with a mounting medium. The sections were then examined under the light microscope.

## Total RNA preparation and qPCR assay
Total RNA was isolated from liver tissue or cultured hepatocytes and homogenized with Trizol (Invitrogen, Carlsbad, CA) and RNeasy kit

(QIAGEN Cat No: 74106). Complementary DNA was synthesized using SuperScript III first-strand (Invitrogen Cat No: 18080051) and quantitative real-time PCR was performed using a 7900HT fast real-time PCR system (Applied Biosystems, Foster City, CA). Primers used for SYBR assay (4367659, Life Technologies, Grand Island, NY) and Taqman PrimeTime® Std qPCR Assays (Integrated DNA Technologies, Inc. Coralville, IA) for fat metabolism genes are listed in the Supplementary Table 2.

## Bulk RNA-sequencing from mouse liver
For each animal, RNA-Seq libraries were constructed using TruSeq RNA Library Preparation Kit (Illumina, San Diego, CA) and sequenced on NovaSeq 6000 System (Illumina) generating roughly 60 million clusters each read as 100 bp paired end reads. Reads were trimmed (trimming of adapters and low quality was doing with cutadapt (switches used for cutadapt -a AGATCGGAAGAGCACACGTCTG AACTCCAGTCA -A AGATCGGAAGAGCGTCGTGTAGGGAAAGAGTGT --overlap 6 -q 20 –minimum-length 25) and aligned to GRCm38 (GENCODE release 27) using STAR (v2.7.3a) 2-pass alignment and quantitated using subread feature counts (v1.6.4) against GEN-CODE v32 gene annotation. Differential expression between four biological replicates for each condition was tested using DESeq2. Within comparisons, mRNAs demonstrating a log2 fold change greater than 0.5 in each direction and adjusted *p*-value under 0.05 were evaluated for GO enrichment using package clusterProfiler (6).

## Serum metabolite measurements
Blood was collected from mice at 22 weeks of age via cardiac puncture immediately after euthanasia, and centrifuged for 10 min at 2000 × *g* to collect serum. Mouse serum lipids were quantified using enzymatic colorimetric assay kits from Fujifilm Wako Pure Chemical Corporation (Osaka, Japan): Wako Cholesterol E for total cholesterol (TC) and Wako L-Type Triglyceride M for triglycerides (TG). Serum non-esterified free fatty acids (NEFA) levels were measured using the mouse Fujifilm Wako HR Series NEFA-HR (2) (Fujifilm Heathcare American Corporation, Lexington, MA # 991-34891). Serum glycerol levels were measured using the glycerol assay kit (Sigma-Aldrich, St. Louis, MO, #MAK117).

## Triglyceride measurement
Small pieces (~ 0.3 g) of liver were taken for triglyceride measurement. A 3:2 mixture of hexane and 2-propanol solvent was added to each sample followed by homogenization with a tissue grinder. The lipid-containing layer was transferred to a new tube. After evaporation, the dried extract was reconstituted with 2-propanol. Triglycerides were measured using the L-type TG M Microtiter Procedure (Wako Diagnostics, Richmond, VA, reagents 461-08992, 461-09092, 464-01601).

## Confocal microscopy

Isolated hepatocytes carrying pEGFP-N1-TFEB were cultured in Nunc Lab-Tek 1.5 Chambered 4 well Coverglass (Thermo Fisher Scientific Inc., Waltham, MA) prior to imaging experiments. Cells were treated with chloroquine, rapamycin, [D-Trp⁸]-γ-MSH or NDP-42 for various times. Live images were acquired during [D-Trp⁸]-γ-MSH or NDP-42 treatment by using Zeiss LSM 880 Airyscan microscopy (Carl Zeiss Inc., Thornwood, NY). Cells were fixed using 4% PFA in PBS at 37 C for 15 min and incubated with DAPI or LysoTracker (Life Technologies, Grand Island, NY) for 5 min for nucleus or lysosome staining, respectively. Image J Fiji (NIH, Bethesda, MD) was used to process and analyze image data.

## Electron microscopy

Mouse livers were immersed and fixed in 2.5% glutaraldehyde made in 0.1 M sodium cacodylate buffer (pH 7.4) for 1 h at room temperature. Tissues were then rinsed in 0.1 M sodium cacodylate buffer. The following processing steps were carried out using the variable wattage Pelco BioWave Pro microwave oven (Ted Pella, Inc., Redding, CA.): post-fixed in 1% osmium tetroxide made in 0.1 M sodium cacodylate buffer, rinsed in double distilled water (DDW), 2% (aq.) uranyl acetate enhancement, DDW rinse, ethanol dehydration series up to 100% ethanol, followed by a Embed-812 resin (Electron Microscopy Sciences, Hatfield, PA.) infiltration series up to 100% resin. The epoxy resin was polymerized for 20 h in an oven set at 60 °C. Ultra-thin sections (90 nm) were prepared on a Leica EM UC7 ultramicrotome. Thin sections were picked up and placed on 200-mesh copper grids (Electron Microscopy Sciences, Hatfield, PA) and post-stained with uranyl acetate and lead citrate. Imaging was accomplished using a JEOL-1400 Transmission Electron Microscope operating at 80 kV and images were acquired on a Gatan UltraScan 1000XP camera with the assistance of the NICHD Microscopy & Imaging Core.

## Droplet digital PCR (ddPCR)

Liver tissue was collected from $Mc3r^{+/+}$ and $Mc3r^{TB/TB}$ mice at ZT2 and ZT12 and RNA was isolated as previously described. RNA (diluted to 70 ng/uL) was DNAse treated and incubated following the iScript gDNA Clear Synthesis Kit protocol with adaptations for ddPCR application. Following DNase treatment, extracted RNA was used as a template to synthesize cDNA using the cDNA Synthesis Kit (Bio Rad: 1725035). DNase-treated RNA samples were treated separately with both iScript reverse transcriptase supermix and No-RT control and reverse transcribed PCR reaction. NRT samples were used to account for background genomic DNA for each replicate. ddPCR was performed on each RT and NRT sample, using beta-actin as a control gene for comparison, according to the ddPCR Supermix for probes (no dUTP) protocol (Bio Rad: #1863024). Probes were obtained from Applied Biosystems for beta-actin (FAM) (Assay ID: Mm02619580_g1) and $Mc3r$ (FAM) (Assay ID: Mm00434876_s1). Droplets were generated using the Automated Droplet Generator (Bio Rad: #1864101) and PCR was performed to amplify DNA within each droplet (1. 94 °C for 10 min, 2. 94 °C for 30 s, 3. 60 °C for 1 min, 4. Step 2–3 repeated 40 times, 5. 98 °C for 10 min, 6. 4 °C hold). The amplified droplets were read using the QX200 Droplet Reader (Bio Rad: #1864003) and analyzed using QuantaSoft to determine $Mc3r$ expression relative to beta-actin expression.

## Plasma γ-MSH measurement

Plasma samples were collected using EDTA and apoprotein as an anticoagulant and protease inhibitor from fed or 24 h fasted mice at 9 AM and 9 PM and were acidified by adding equal volume of buffer A (Phoenix Pharmaceuticals Cat. No. RK-BA-1). Samples were then mixed and spun at $17,000 \times g$ for 20 min (4 °C). A C-18 SEP-COLUMN (Phoenix Pharmaceuticals RK-SEPCOL-1) was equilibrated by washing with 1 mL buffer B once (Phoenix Pharmaceuticals RK-BB-1) and 3 mL of buffer A three times. The acidified plasma solution was loaded onto the C-18 SEP-Column then washed with 3 mL buffer A twice before eluting in 750 µL buffer B. The eluted samples were then concentrated by speed vacuum and lyophilized. The samples were resuspended in equal volume buffer A before γ-MSH plasma concentration was measured by ELISA (EK-043-01).

## In vivo γ-MSH administration

Singly housed 12-week-old C57BL/6J mice in the fed state were injected intraperitoneally with saline or [D-TRP⁸] γ-MSH (Phoenix Pharmaceuticals 043-10) at a dosage of 200 µg/kg bodyweight and then sacrificed at 30, 60, 90, or 120 min post-injection. Liver samples were collected, and proteins were freshly prepared. Hepatic expression of LC3I/II was examined via western blot analysis. Primary LC3 antibodies were diluted 1:1000 and β-actin 1:5000. The absolute absorbance of LC3II and β-actin were quantified using Amersham software and the values of LC3II (corrected for β-actin) were used to determine activation of autophagy.

## Statistics analysis and Reproducibility

Sample sizes were based on prior animal studies, suggesting meaningful results from 5–20 animals per group. We mentioned exact independent observations in legends and exact $p$ values in data file. We performed three time independent experiments for representative experiments, such as micrographs for tissue histology and western blots. For electron microscopy and super-resolution confocal images, two independent experiments had the analysis for subcellular analysis using multiple images. Data are expressed as mean ± S.E.M. unless otherwise indicated. Throughout the study, $*p < 0.05$; $**p < 0.01$; $***p < 0.001$. GraphPad Prism 9.2.0 software (GraphPad Software, San Diego, CA) was used for Student's $t$-tests (two-tailed), one-way ANOVA followed by Tukey's HSD test or Fisher's LSD test, and two-way analysis of variance followed by Tukey's HSD or Bonferroni/Dunn's post-hoc test or Friedman's test by Dunn's paired analysis. IBM SPSS 27 software (Armonk, New York) was used to perform analysis of covariance for serum data adjusted for fat mass. Energy expenditure data was examined using analysis of covariance (ANCOVA) adjusted for body weight, lean & fat mass, and lean mass, and post hoc tests for estimated marginal means were calculated using SPSS 28.0.2. Differences were considered significant at $p = 0.05$. Data met assumptions of the statistical tests, including requirements for similar variance across groups.

## Reporting summary

Further information on research design is available in the Nature Portfolio Reporting Summary linked to this article.

## Data availability

The datasets analyzed for the current study are deposited at https://doi.org/10.6084/m9.figshare.23900223. RNAseq data are deposited as PRJNA1180604 [https://www.ncbi.nlm.nih.gov/bioproject/1180604]. Source data are provided with this paper.

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

## Acknowledgements

We thank Drs. Noboru Mizushima (RIKEN Bio http://www.brc.riken.jp/lab/animal/en/dist.shtml) and Sergey Leikin (Cell Biology and Metabolism Program, NICHD) for the kind gift of GFP-LC3 transgenic mice. We thank Dr. Jennifer Lippincott-Schwartz (Janelia Research Campus) for providing resources for imaging analysis. We also thank the NICHD Microscopy and Imaging Core for their assistance with performing transmission electron microscopy. We acknowledge the NIDDK Mouse Metabolism Core for their assistance performing mouse indirect calorimetric studies and serum-free fatty acid assay. We thank Oksana Gavrilova for review and help in interpreting energy expenditure results. We acknowledge the Molecular Genomics Core at NICHD for Bulk RNAseq experiments. We gratefully acknowledge the assistance of the NICHD animal care facility for the research support of animal breeding and housing. This research was supported by the intramural research program of the Eunice Kennedy Shriver National Institute of Child Health and Human Development (NICHD), grant ZIAHD00641 (to J.A.Y.) with supplemental funding from an NICHD Early Career Investigator Award (to T.P.P.), National Institute of Diabetes and Digestive and Kidney Diseases (NIDDK) grant 1R01DK124504 (to C.H.L.) with supplemental support from an NIH Bench-to-Bedside award (to J.A.Y.) made possible by the NIH Office of Clinical Research. The NICHD and co-funders had no role in the design and conduct of the study, collection, management, analysis, or interpretation of the data, preparation of the manuscript for publication, or decision to submit the manuscript for publication. NICHD did review the manuscript and approve its submission. T.P.P, J.Y.J., A.Y.S., and J.A.Y. had access to all the data in the study and take full responsibility for the integrity of the data and the accuracy of the data analysis. The opinions and assertions expressed herein are those of the authors and are not to be construed as reflecting the views of the National Institutes of Health or the US Department of Health and Human Services.

## Author contributions

T.P.P., J.Y.J., A.Y.S and J.A.Y. designed, carried out and interpreted the experiments. T.P.P., J.Y.J., A.Y.S and J.A.Y. wrote the manuscript. T.P.P, A.Y.S. and J.A.Y. prepared figures. T.P.P., N.J.L. carried out the liver-specific recovery study, and T.P.P. performed the cell fractionation and cell biology experiments. A.M.W., D.E., J.C., N.T., E.K.A., D.B.G., S.M.J. and P.P. helped with cell and animal experiments. T.C.D. acquired energy expenditure data. M.E. and C.H.L. synthesized MC3R specific agonist, NDP-42 compound. A.C. helped perform RNAseq analysis. A.W. performed serum lipid measurements. All authors provided critical review of the manuscript and gave final approval of the submitted version.

## Funding

## Competing interests

J.A.Y. receives grant support unrelated to this article for pharmacotherapy trials for human obesity from Hikma Pharmaceuticals, Inc., Soleno Therapeutics, Inc., and Rhythm Pharmaceuticals, Inc., and reagents (anti-activin receptor antibodies) from Versanis Bio for mouse studies. The remaining authors declare no competing interests.
