## [Transparent Peer Review file · Nature Communications]

Melanocortin 3 receptor regulates hepatic autophagy and systemic adiposity

Corresponding Author: Dr Jack Yanovski

Version 0:

Reviewer comments:

Reviewer #1

(Remarks to the Author)

In this study, the authors describe a previously unidentified role of melanocortin-3 receptor (MC3R) expressed by the liver. The authors demonstrate that fat is increased, and that autophagy is dysfunctional in the liver Mc3r-deficient mouse models. Liver fat and dysfunctional autophagy was reversed by liver-specific re-expression of Mc3r. The authors further suggests that gamma-MSH stimulates MC3R and TFEB. Overall, this study revealed a novel function of MC3R expressed by the liver and the data are generally well-presented. However, there are several issues that need to be addressed to make this study more convincing.

Major

1. Figs. 1c, 3g, and 3h, the authors report a big obesity phenotype in Mc3r-deficiency mice. However, previous studies reported only mild obesity in mouse models (references 13-15). Were the Mc3r-deficient mice hyperphagic? It is necessary to explain the discrepancy between independent studies.
2. Figs. 3g and 3h, it is remarkable that liver-specific re-expression of MC3R reverses much of the obesity phenotype. While liver is a large organ, the reversal of body weight is not likely to be entirely explained by decreased fat in the liver. The authors need to provide more evidence, such as muscle mass, fat mass, food intake, energy expenditure, to explain such a big phenotype.
3. The authors suggested gamma-MSH as an endogenous agonist to stimulate MC3R to result in autophagy in the liver. However, it is not clear how gamma-MSH may promote starvation-induced autophagy via MC3R expressed by the liver. While starvation induces LC3II, it is unlikely due to the stimulation of MC3R by gamma-MSH because POMC neuronal activity is low during starvation. Unless supported by further evidence, it is hard to believe that gamma-MSH works in vivo for MC3R-dependent autophagy in the liver.

Minor

1. Introduction, it may be better to add a paragraph to briefly describe what the authors in the present study.
2. Fig. 1d shows a big increase of adiposity in Mc3r-deficient mice. It would be necessary to show H&E images of liver and some white fat tissues to make this data more convincing.
3. Figs. 2d and 2e, what is the authors' interpretation on the higher liver LC3II observed in Mc3r-deficient mice in fed conditions? If LC3II expression is already high in fed condition, does it mean active autophagy even before starvation?
4. Figs. 1a, 2e and suppl Fig. 3b, statistical analysis should be done between all experimental groups.
5. Figs. 2h and 2i, what happens after starvation?
6. Suppl Fig. 3e and 3f, quantification does not seem to summarize the western blot data. Also, please perform statistical analysis for the data in 3f.

7. Fig. 4a, gamma-MSH treatment in the center panels should be repeated in Mc3r-deficient mice to see if observed results are indeed MC3R-dependent.

8. Fig. 4j, it appears that the absence of MC3R itself increases TFEB expression. What is the authors' explanation on this result?

Reviewer #2

(Remarks to the Author)

This work by Patel and Jun et al investigates the non-canonical role of liver melanocortin 3 receptor in the control of hepatic autophagy and systemic adiposity. Using multiple loss of function models of Mc3r, the authors demonstrate that loss of Mc3r leads to increased hepatic triglyceride accumulation that correlates with defective autophagy. A series of nicely performed studies show that activation of Mc3r is sufficient to induce hepatic autophagy via the TFEB pathway. Lastly, liver-expression of Mc3r partially restores hepatic autophagy and is sufficient to reduce body weight and improve insulin resistance. The studies are nicely performed and the notion that Mc3r plays a role in hepatocytes and contributes to known body weight effects of Mc3r are quite exciting. This would lead to a significant advance in the field and further our understanding of how variants in Mc3r are associated with the regulation of body weight. With that being said, there are several additional experiments that should be performed to further strengthen and validate the major conclusions of this study.

1) The authors took advantage of global Mc3r knockout animals and relied on re-expression of Mc3r in hepatocytes in the knockout animals to implicate an autonomous effect of Mc3r in liver. Ideally, a liver specific knockout of Mc3r alone would further define the specific contribution of hepatic Mc3r. Nevertheless, protein expression and staining of Mc3r should be performed to validate the mRNA effects in Fig 1 to further validate Mc3r expression (at the protein level) in hepatic tissue.

2) A major unanswered question is why does re-expression of liver Mc3r reduce body weight is unanswered. A detailed assessment of energy expenditure, food intake and locomotor activity should be performed to point to a mechanism underlying how restoring Mc3r in hepatocytes can control system body weight.

3) The changes in fatty liver, insulin resistance and glucose levels are quite dramatic. Are these driven solely by the weight gain or is Mc3r affecting hepatic insulin sensitivity independent of body weight. Pair feeding studies would be one way to disentangle these effects.

3) Defective autophagy is suggested to drive the changes in hepatic steatosis and point to a role for TFEB. The authors performed a series of in vivo and in vitro studies that does support this. However, without functional restoration of autophagy (potential by altering TFEB), it is difficult to determine if this is indeed the sole driver of the phenotype. Moreover, direct measurements of fatty acid esterification, oxidation and lipogenesis are necessary to further confirm that autophagy abnormalities are driving hepatic steatosis.

Version 1:

Reviewer comments:

Reviewer #1

(Remarks to the Author)

In the revised version of manuscript, the authors performed additional experiments to address concerns raised by this and the other reviewer. The manuscript has been improved but there are still several unresolved issues as specified below.

1. It is still unclear how obesity observed in Mc3r-deficient mice are reversed by the re-expression of Mc3r in the liver only. As noted by the authors in Figs. 1a and 1b, expression level of Mc3r in the liver is quite low. While this level of Mc3r may have a role in the liver, as the authors demonstrated, it is not very likely that this level of Mc3r can affect body weight and whole-body adiposity when Mc3r is absent everywhere else. The authors showed additional data in this revision, but the most important data – changes in energy expenditure – to explain the observed phenotype is missing. One may readily understand that re-expression of Mc3r in the liver rescues adiposity and autophagy of the liver, but if the authors want to claim that body weight and whole-body adiposity are reversed by the re-expression of Mc3r in the liver, the mechanism for this change is key to support it. Therefore, it is necessary to examine what happens to the energy expenditure (for example, as determined by indirect calorimetry) and locomotion. It may also be a good idea to look at BAT.

2. Supplemental Figure 3d, the authors stated that daily food intake was not changed in the Mc3r-deficient mice, but that feeding efficiency tended to be increased when both are not significantly different between control and Mc3r-deficient mice (page 9, lines 202-206). Please revise this part of the manuscript to accurately describe the findings.

3. Figure 1a, Mc3r expression levels are not significantly different at ZT12 and ZT2. Therefore, the authors' interpretation that Mc3r expression has diurnal variation (page 5, lines 95-99) is not correct unless they perform more experiments to obtain statistical significance.

4. Figures 1g-1h and Supplemental Figures 1a-1b, why do Mc3r-deficient livers have higher numbers of lipid droplet numbers in Figures 1g-1h, but not in Supplemental Figures 1a-1b? Were the experiments in Figures 1g-1h performed in fed

conditions? If so, it should be mentioned in the text and/or figure legends.

Reviewer #2

(Remarks to the Author)

Thank you to the authors for the additional data and textual clarifications to improve the manuscript. One significant concern remains unaddressed. As also noted by reviewer 1, the ability of re-expression of Mc3r to reverse the obesity phenotype is very surprising (and a main crux of the paper). This phenotype should be supported by complementary analysis of energy balance. It is appreciated that additional cohorts of mice are required that will take several months to generate but these are important physiological studies that (in my opinion) are necessary to support the conclusions of the main paper.

Version 2:

Reviewer comments:

Reviewer #1

(Remarks to the Author)

The authors performed additional experiments to assess energy expenditure, locomotor activity, and BAT phenotypes in this round of revision. In the new figure 4, the authors reported small but significant decreases of energy intake, total energy expenditure, and oxygen consumption in the Mc3r^{TB/TB} mice compared to Mc3r^{+/+} mice. These phenotypes were not reversed in Mc3r^{Hep/Hep} mice. Therefore, it still remains unanswered how re-expression of MC3R only in the liver (Mc3r^{Hep/Hep} mice) reverses body weight as reported in figures 3i and 3j. The authors claim in discussion (page 16, lines 401-403) that "However, it remains possible that the small differences in energy expenditures as observed here might contribute to the more-normal body composition of Mc3r^{Hep/Hep}.", but it is not clear if this small and insignificant differences of energy expenditure may result in such a robust restoration phenotype of body weight.

Reviewer #2

(Remarks to the Author)

The addition of new calorimetry data are a nice addition to the overall manuscript and necessary to understand the impact of Mcr3 on energy homeostasis

Version 3:

Reviewer comments:

Reviewer #1

(Remarks to the Author)

The authors have satisfactorily addressed all of my previous concerns. Normalized energy expenditure in the liver Mc3r re-expression mice explains the body weight phenotype. The Seahorse and RNAseq data are also helpful.

Reviewer #1 (Remarks to the Author):

In this study, the authors describe a previously unidentified role of melanocortin-3 receptor (MC3R) expressed by the liver. The authors demonstrate that fat is increased and that autophagy is dysfunctional in the liver Mc3r-deficient mouse models. Liver fat and dysfunctional autophagy was reversed by liver-specific re-expression of Mc3r. The authors further suggests that gamma-MSH stimulates MC3R and TFEB. Overall, this study revealed a novel function of MC3R expressed by the liver and the data are generally well-presented. However, there are several issues that need to be addressed to make this study more convincing.

Major

1. Figs. 1c, 3g, and 3h, the authors report a big obesity phenotype in Mc3r-deficiency mice. However, previous studies reported only mild obesity in mouse models (references 13-15). Were the Mc3r-deficient mice hyperphagic? It is necessary to explain the discrepancy between independent studies.

Response: *We completely agree with the reviewer's comment and confirm that Mc3r deficient mice have a moderate form of obesity, particularly at younger ages. It is known that global Mc3r deficiency is a relatively mild late-onset form of obesity, known to increase body weight via a significant increase in fat mass and reduction in lean mass (Begrache et al., 2011; Renquist et al., 2012). We utilized the transcriptionally blocked global knockout Mc3r^{TB/TB} mouse model, first developed by Butler's group, who showed no change in total body weight in younger-aged mice (8 weeks, chow), and a mild increase (trend, $p = 0.06$) in body weight in the adult (12 weeks, chow) mouse obesity phenotype due to an increase in fat mass (Begrache et al., 2011). In this study, we observed a significant obesity phenotype with body weight differences that started diverging only at 16 weeks of age and continued up to 22 weeks in our Mc3r^{TB/TB} deficiency mice (Female: 31.0 ± 4.53 , $P < 0.0001$; Male: 39.3 ± 4.26 , $P < 0.002$) mice compared to the wild-type control (Female: 21.8 ± 1.63 ; Male: 29.8 ± 6.01). Importantly, our data are in line with previous works reporting WT 26.44 ± 0.49 vs. 30.58 ± 1.27 Mc3r^{-/-}; ** $P < 0.01$ in the Mc3r^{-/-} global knockout mouse model with a different background (Renquist et al., 2012). Given these observations, we monitored until 22 weeks of age to see the development of the obesity phenotype, as reported in Figure 3 a-b, g-h.*

It is well known that Mc4r^{-/-} mice exhibits severe obesity and are 50% greater in body weight than heterozygotes with a single nonfunctional allele (Mc4r^{+/-}) (Collet et al., 2017; Hatoum et al., 2012). We now include in the report data collected contemporaneously from heterozygous Mc4r^{+/-} mice and show the body weight is quite comparable to that seen in the transcriptionally blocked global knockout Mc3r^{TB/TB} mice. To better represent our findings, we now show our mouse model Mc3r^{TB/TB} body weight curve compared to the Mc4r^{+/-} mice in revised Supplementary Figs 3a and 3b. These supporting data showed that Mc3r^{TB/TB} homozygotes gain

similar body weight to *Mc4r* heterozygotes, confirming that *Mc3r* insufficiency causes a relatively moderate form of obesity.

*Obesity of $Mc3r^{-/-}$ mice is independent of hyperphagia, and it is also believed to result from altered peripheral metabolism in addition to changes in the known central role of *Mc3r* (Begrache et al., 2011). The obesity of *Mc3r* deficient mice, $Mc3r^{TB/TB}$, was present in the absence of food intake differences between $Mc3r^{TB/TB}$ and respective control groups (Begrache et al., 2011).*

*In line with these studies, food intake did not differ between groups in previous loss of function mutations or genetic deletion of *Mc3r* mice in both chow and high-fat diet-fed conditions after adjusting separately for both fat and fat + lean mass (Lee et al., 2016). We have expanded on this information in the introduction (Page 3, Line 51-53, 59-60) and results (Page 9, Line 198-206) sections in the main text, respectively:*

Introduction:

*“In mice, global *Mc3r* deficiency obesity is a mild, late-onset form of obesity, that increases body weight moderately because it increases fat mass while reducing in lean mass (Begrache et al., 2011; Renquist et al., 2012). Previous studies have demonstrated neither hyperphagia nor global hypometabolism in *Mc3r* deficiency, leaving the mechanism of increased adiposity incompletely determined (Ghamari-Langroudi et al., 2018; Renquist et al., 2012).”*

Results:

*“We also compared results to contemporaneously studied $Mc4r^{+/-}$ mice, the far less severely affected heterozygotes of the severe monogenetic obesity $Mc4r^{-/-}$ mouse (Supplementary Fig. 3a and 3b). Our data finding $Mc3r^{TB/TB}$ homozygotes gain similar body weight compared to *Mc4r* heterozygotes confirm that *Mc3r* insufficiency is a moderate form of obesity. Consistent with previous reports, daily food intake was not significantly increased in *Mc3r* deficient mice (Supplementary Fig. 3c), even after adjusting for body weights and lean mass, but there was a nonsignificant trend towards greater feeding efficiency in $Mc3r^{TB/TB}$ that was ameliorated by liver specific *Mc3r* recovery (Supplementary Fig. 3d).”*

2. Figs. 3g and 3h, it is remarkable that liver-specific re-expression of MC3R reverses much of the obesity phenotype. While liver is a large organ, the reversal of body weight is not likely to be entirely explained by decreased fat in the liver. The authors need to provide more evidence, such as muscle mass, fat mass, food intake, and energy expenditure, to explain such a big phenotype.

Response: *We agree with reviewer and have now provided new data for percentage of Lean Mass (Fig. 3b), Food Intake (Supplementary Fig. 3b) and Feeding efficiency (Supplementary Fig. 3c), where we now report that our *Mc3r* deficient obesity mouse model showed normal food intake and feeding efficiency. We have added these new results and text describing previous data for both food monitoring and energy expenditure, specifically describing both *Mc3r* deficient mouse models from our and independent studies in the introduction (Page 3-4, Line 61-69), results (Page 9, Line 202-206), and discussion section (Page 14-15, Line 337-348).*

Unfortunately, it will take another year to generate energy expenditure data from the important mouse genotypes in the current manuscript and therefore we have noted the need for such studies in the Discussion rather than supplying these additional data.

Introduction:

“In addition to normophagia, these mice have normal total energy expenditures (Butler et al., 2000; Sutton et al., 2006; Zhang et al., 2005). Furthermore, resting energy expenditure is not significantly affected in Mc3r deficient Mc3r^{TB/TB} mice, compared to wild-type mice (Begrache et al., 2011). There is, however, some evidence for a physical activity phenotype, with reduced locomotion in Mc3r^{TB/TB} mice during the dark period (Begrache et al., 2011). Changes in physical activity could impact total daily energy expenditure; however, this effect does not translate into altered 24-h energy expenditure (Begrache et al., 2011). In loss-of-function humanized double mutant mice, skeletal muscle weight data are also consistent with the lack of differences in energy expenditures (Lee et al., 2016). Overall, in Mc3r deficient mice, efficiency of energy utilization appears increased, and there is no marked energy expenditure phenotype.”

Results:

“Consistent with previous reports, daily food intake was not significantly increased in Mc3r deficient mice (Supplementary Fig. 3c), even after adjusting for body weights and lean mass, but there was a nonsignificant trend towards greater feeding efficiency in Mc3r^{TB/TB} that was ameliorated by liver specific Mc3r recovery (Supplementary Fig. 3d).”

Discussion:

“However, we saw that the recovery of liver specific Mc3r did not affect food intake and feeding efficiency (as indicated in Supplementary Fig. 3). Since, total daily energy expenditure is not significantly altered in MC3R-deficient mice (Butler et al., 2000; Sutton et al., 2006; Zhang et al., 2005), we did not anticipate there would be a hepatic-specific role of Mc3r in regulating energy expenditure. Recently, it was found that in the lateral hypothalamic area (LHA) and ventromedial hypothalamic area (VMH) Mc3r play crucial roles in regulating feeding, rheostasis, body weight maintenance, and metabolism (Dunigan et al., 2021; Pei et al., 2019). Energy expenditure was lower in Mc3r^{LHA} ablation mice without a change in food intake or fast–refeeding food intake and accrual of body fat mass. These recent results confirm that Mc3r in major hypothalamic Mc3r-neurons may impact energy expenditure and locomotion (Dunigan et al., 2021; Pei et al., 2019). Thus, further investigation is required to elucidate if changes in energy expenditure and physical activity contribute to the phenotypic improvements noted in hepatic recovery mice.”

3. The authors suggested gamma-MSH as an endogenous agonist to stimulate MC3R to result in autophagy in the liver. However, it is not clear how gamma-MSH may promote starvation-induced autophagy via MC3R expressed by the liver. While starvation induces LC3II, it is unlikely due to the stimulation of MC3R by gamma-MSH because POMC neuronal activity is low during starvation. Unless supported by further evidence, it is hard to believe that gamma-MSH works in vivo for MC3R-dependent autophagy in the liver.

Response: We apologize for any confusion and have carefully examined the manuscript to ensure there are no sentences suggesting this was the case. We did not fast the mice prior to in vivo injection of the MC3R agonist, gamma MSH and have not indicated there are defects in fasting-induced autophagy in Mc3r deficiency. Indeed, it appears that peripheral ACTH concentrations (and therefore likely peripheral gamma-MSH concentrations) are reduced in mice during food restriction or fasting (Han et al., 1998; Lee et al., 2008), There are also published data demonstrating that activation of the CNS MC3R leads to increased energy intake (Lee et al., 2008; Marks et al., 2006; Sweeney et al., 2021); thus reduced food intake is not part of the response to MC3R agonist administration in vivo. We agree with the reviewer that the MC3R pathway stimulates autophagy through a mechanism that is also involved in the starvation pathway, but we can provide no definitive data for MC3R involvement in the starvation-induced autophagy response. We show that in vitro and in vivo activation of MC3R can modulate hepatic autophagy; liver LC3II was examined in Mc3r^{+/+} mice (fed) after intraperitoneal administration of [D-Trp8]- γ -MSH compared to the saline-injected control group (Fig. 5h and supplementary Fig. 4e and 4f). In addition to in vivo data, biochemical, and imaging results in isolated primary hepatocytes confirm increased LC3II upon [D-Trp8]- γ -MSH exposure in a time-dependent manner (Fig. 5c). The complete Figure 5 show how MC3R agonist stimulates LC3II-mediated autophagy in vivo and isolated primary hepatocytes (Fig. 5a-i). We have added the following text to the manuscript's Discussion (Page 14, Line 331-335):

“These data do not indicate that the MC3R is involved in the stimulation of hepatic autophagy by fasting or starvation. Indeed, it appears that peripheral ACTH concentrations (and therefore likely peripheral gamma-MSH concentrations) are reduced in mice during prolonged food restriction or fasting (Han et al., 1998; Lee et al., 2008).”

Minor

1. Introduction, it may be better to add a paragraph to briefly describe what the authors in the present study.

Response: We thank the review for their comment. We added a paragraph to briefly describe the present study: (Page 4, Line 78-85)

“However, how Mc3r acts in the liver to affect systemic adiposity has yet to be fully explained. Given that proper hepatic autophagy is needed for active energy redistribution, stimulating both formation and recycling of cellular lipid-droplets (LDs) (Liu and Czaja, 2013; Singh, 2010; Singh et al., 2009) we aimed to investigate the liver-specific role of Mc3r in LD-autophagy, finding that activation of MC3R signaling induces autophagy in wild-type but not Mc3r-deficient mice, and that autophagy flux is dysregulated in Mc3r deficiency, with defective lysosomal turnover. Further, our data show liver specific Mc3r rescue in the context of global Mc3r deficiency restored liver autophagy and reduced adiposity.”

2. Fig. 1d shows a big increase of adiposity in Mc3r-deficient mice. It would be necessary to show H&E images of liver and some white fat tissues to make this data more convincing.

Response: We thank the reviewer for their feedback and have made updates accordingly. Our latest additions include H&E staining and Oil Red O image data for the liver (Fig. 4a-b), as well as H&E staining for epididymal white adipose tissue (eWAT) for the $Mc3r^{+/+}$, $Mc3r^{TB/TB}$ and $Mc3r^{Hep/Hep}$ in Supplementary Fig. 3e.

3. Figs. 2d and 2e, what is the authors' interpretation on the higher liver LC3II observed in $Mc3r$ -deficient mice in fed conditions? If LC3II expression is already high in fed condition, does it mean active autophagy even before starvation?

Response: Thank you for this excellent observation. Our study suggests that lysosomal turnover is not normally functioning due to defective $Mc3r$, leading to LC3II accumulation. Despite higher levels of LC3II observed in $Mc3r$ -deficient hepatocytes, this is mainly due to a defect in autophagosome degradation and recycling of LC3 for a new round of the autophagy process, which indicates a lack of active autophagy. We experimentally supported this finding by blocking autophagic flux (AP degradation) with chloroquine, which increased LC3II and p62 degradation in wild-type primary hepatocytes, not in $Mc3r$ deficient cells, demonstrating defective AP turnover kinetics (AP flux) under fed or serum-starved conditions (Figs 2f and g). These findings are described in results in detail (Page 7, Line 149-156).

"We found that under fed conditions (when mice maintain basal autophagy activity in the liver), both $Mc3r^{TB/TB}$ and $MC3R^{hDM/hDM}$ mice exhibited much greater basal LC3II protein levels in liver compared to control mice (Figs 2d and 2e) due to defective $Mc3r$, leading to LC3II accumulation. As expected, under fasting conditions, there was significant induction of liver LC3II levels in control mice (Figs 2d and e), confirming our fasting condition enhances hepatic autophagy. However, the similarly fasted $Mc3r^{TB/TB}$ and $MC3R^{hDM/hDM}$ mice failed to increase liver LC3II (Figs 2d and e), suggesting impaired autophagic regulation in $Mc3r$ -deficient livers."

The entire section related to "Defective autophagosome turnover in $MC3R$ deficient liver" (pages 7-8, lines 143-186) is also relevant to this question.

In Discussion, the following sentences are relevant to this question:

"We show in two mouse models of $MC3R$ insufficiency that autophagic flux is disrupted in a unique manner"

"A specific $MC3R$ agonist, NDP-42, activated hepatic LC3II in $Mc3r^{+/+}$ hepatocytes but did not further activate LC3II in $Mc3r^{TB/TB}$, which showed blunted autophagosome-lysosome docking and LD-clearance, leading to defective lipid autophagy, thus confirming the role of $MC3R$ pathway in hepatocytes"

"Our observations from $Mc3r^{TB/TB}$ or $MC3R^{hDM/hDM}$ livers lead us to hypothesize that the $MC3R$ pathway functions specifically in determining lysosomal-AP interaction and degradation processes, based on the: 1) prevalent AP accumulation with luminal compartments containing undegraded organellar membranes (Figs 2a-c and h-i), 2) impaired AP flux and p62 clearance

upon serum starvation (Figs 2f, g and Supplementary Fig. 1c), and 3) increased basal LC3II in both hepatic tissues and isolated hepatocytes (Figs 2d, e). In addition, phospho-4E-BP1 degradation confirms the LC3II activation and change in MTORC1 might be possible upstream autophagy mediators (Supplementary Fig. 6a). Moreover, the findings of MC3R-dependent nuclear TFEB allocation (Figs 5j) with subsequent autophagy and lysosomal gene activations (Fig 5i) further support MC3R's specific roles in the regulation of autophagy.”

4. Figs. 1a, 2e and supple Fig. 3b, statistical analysis should be done between all experimental groups.

Response: We have revised figures to now include statistical analysis between all experimental groups in Figs. 1a, 2e and *Supplementary Fig. 3b*, whose data are now in *Supplementary Fig. 4b*.

5. Figs. 2h and 2i, what happens after starvation?

Response: *Thanks for reviewer's comment. We have shown that fasting in the wild-type liver leads to activation of LC3II and hence the autophagosome, while Mc3r deficient hepatocytes showed no future activation with the starvation of LC3II and autophagosome structure as shown in figure 2a-e. Biochemical data from the Mc3r deficient hepatocytes showed that LC3II was not further stimulated by starvation (Fig. 2f), unlike wild-type cells. It is well established that after starvation, primary hepatocytes in the wild-type group show similar activation of autophagy to rapamycin stimulation, as shown in Fig. 5a. In Figs. 2h and 2i, we show a basal autophagy defect in Mc3r deficient hepatocytes without stimulation. Starvation will form similar LC3 aggregates, as formed in control hepatocytes when AP degradation was blocked by chloroquine (Supplementary Fig. 1d), and we have described this in results section in detail (Page 7, Line 149-156; Page 8, Line 180-186).*

“We found that under fed conditions (when mice maintain basal autophagy activity in the livers), both $Mc3r^{TB/TB}$ and $MC3R^{hDM/hDM}$ mice exhibited much greater basal LC3II protein levels in liver compared to control mice (Figs 2d and 2e) due to defective Mc3r, leading to LC3II accumulation. As expected, under fasting conditions, there was significant induction of liver LC3II levels in control mice (Figs 2d and e), confirming our fasting condition enhances hepatic autophagy. However, the similarly fasted $Mc3r^{TB/TB}$ and $MC3R^{hDM/hDM}$ mice failed to increase liver LC3II (Figs 2d and e), suggesting impaired autophagic regulation in Mc3r-deficient livers.”

“Conversely, hepatocytes from $Mc3r^{TB/TB}$ or $MC3R^{hDM/hDM}$ mice displayed many GFP-LC3 vesicles that were aggregated even without serum starvation (Fig. 2h, Inlay). Similar LC3 aggregates were formed in control hepatocytes when AP degradation was blocked by chloroquine (Supplementary Fig. 1d). Additional 3D reconstruction of GFP-LC3 structures further revealed augmented AP volumes in Mc3r-insufficient hepatocytes compared to control cells (Fig. 2i). These results clearly support impaired AP turnover caused by Mc3r deficiency in mouse liver.”

6. Suppl Fig. 3e and 3f, quantification does not seem to summarize the western blot data. Also, please perform statistical analysis for the data in 3f.

Response: We thank the Reviewer for noting the need to revise these figures. *The Suppl Fig 3 is now Suppl Fig 4. We appreciate the reviewer comments, and we now show the correct representative version of the western blot quantification for in vivo gamma MSH injection at 30-, 60- and 120-mins western blot data and included statistical analysis in supplementary Fig. 4f.*

7. Fig. 4a, gamma-MSH treatment in the center panels should be repeated in Mc3r-deficient mice to see if observed results are indeed MC3R-dependent.

Response: *We appreciate the reviewer's comments and performed live cell imaging time lapsed studies in Mc3r-deficient mice. We now provide a new data set showing that treatment with a known MC3R agonist, gamma MSH, along with wild-type control activates the LC3-mediated autophagy pathway in an MC3R-dependent manner in Supplementary Fig. 5.*

8. Fig. 4j, it appears that the absence of MC3R itself increases TFEB expression. What is the authors' explanation on this result?

Response: *In primary hepatocytes nuclear fraction, deficiency of Mc3r resulted in accumulated TFEB in the nucleus and failed to promote further stimulation, unlike wild-type control cells. When TFEB de-phosphorylates, it is translocated to the nucleus and triggers the activation of genes that are responsible for the formation of lysosomes, autophagy (specifically lipophagy), and the exocytosis of lysosomes. In case of persistent cellular stress, TFEB is unable to recycle back to the cytoplasm, and uncontrolled activation can have deleterious effects on cellular functions, which can contribute to the development of disease state (Franco-Juarez et al., 2022). In line with this, our results suggest that accumulated basal nuclear TFEB in Mc3r deficient hepatocytes are unable to recycle back to the cytoplasm and translocate to the nucleus with further stimulation either by starvation or gamma MSH for the normal cellular function of autophagy. We have now included the total TFEB expression data for whole-cell protein lysate for the Mc3r deficient and hepatic rescue groups, which showed no significant changes compared to the wild-type groups (refer to Supplementary Fig. 2b).*

Reviewer #2 (Remarks to the Author):

This work by Patel and Jun et al investigates the non-canonical role of liver melanocortin 3 receptor in the control of hepatic autophagy and systemic adiposity. Using multiple loss-of-function models of Mc3r, the authors demonstrate that loss of Mc3r leads to increased hepatic triglyceride accumulation that correlates with defective autophagy. A series of nicely performed studies show that activation of Mc3r is sufficient to induce hepatic autophagy via the TFEB pathway. Lastly, liver expression of Mc3r partially restores hepatic autophagy and is sufficient to reduce body weight and improve insulin resistance. The studies are nicely performed and the notion that Mc3r plays a role in hepatocytes and contributes to known body weight effects of Mc3r are quite exciting. This would lead to a significant advance in the field and further our

understanding of how variants in Mc3r are associated with the regulation of body weight. With that being said, there are several additional experiments that should be performed to further strengthen and validate the major conclusions of this study.

1) The authors took advantage of global Mc3r knockout animals and relied on the re-expression of Mc3r in hepatocytes in the knockout animals to implicate an autonomous effect of Mc3r in liver. Ideally, a liver-specific knockout of Mc3r alone would further define the specific contribution of hepatic Mc3r. Nevertheless, protein expression and staining of Mc3r should be performed to validate the mRNA effects in Fig 1 to further validate Mc3r expression (at the protein level) in hepatic tissue.

Response: *These are excellent suggestions. However, to our knowledge, there are no reliable and/or knockout validated antibodies to confirm protein expression using Immunoblotting or IHC. We have tried using multiple commercially available antibodies and even requested antibodies to be synthesized, but we have not been successful in obtaining a reliable Mc3r specific antibody to be employed in our work. Hence, we used sensitive digital droplet PCR (for liver samples collected at different zeitgeber time points) and confirmed our findings using functional studies either by cAMP levels (Supplementary Figure 4b) and functional activation of autophagosome structure of MC3R in primary hepatocytes using MC3R agonists in autophagy to MC3R dependent activation. We thank for the suggestion about using liver-specific knockout mouse model, which we are now in the process of pursuing. This need is now added the text in as a future direction to the discussion section (Page 16, Line 379-381).*

“In order to delve deeper into the impact of MC3R on hepatic lipid droplet autophagy, it will be necessary to utilize a mouse model with a tissue-specific knockout.”

2) A major unanswered question is why does re-expression of liver Mc3r reduce body weight is unanswered. A detailed assessment of energy expenditure, food intake and locomotor activity should be performed to point to a mechanism underlying how restoring Mc3r in hepatocytes can control system body weight.

Response: *We appreciate the reviewer’s comments. We have added data for the assessment of food Intake monitoring (Supplementary Fig. 3c) and feeding efficiency (S. Fig. 3d). Unfortunately, it will take us at least 1 more year to supply energy expenditure data from the relevant mice, which we are also in the process of obtaining to be used in a subsequent manuscript, so we have noted the need for such studies in the Discussion as we cannot supply new data on this issue.*

Mc3r deficiency is a mild late-onset form of obesity, known to increase body weight via a significant increase in fat mass and reduction in lean mass (Begrache et al., 2011; Renquist et al., 2012). The Mc3r deficient obesity mouse model showed normal food intake and energy expenditure. Additionally, multiple investigations have shown that Mc3r deficient obesity mouse models did not show significant differences in total energy expenditure using indirect calorimetry when compared to the TB global knockout mouse model. Furthermore, resting energy

expenditure was not significantly affected in *Mc3r^{TB/TB}* compared to WT without altering food intake. Locomotion was reduced in *Mc3r^{TB/TB}* mice during the dark period and changes in physical activity could impact energy expenditure, however, this effect did not translate into 24-h energy expenditure (Begriche et al., 2011).

*In our previous work (Lee et al., 2016), we measured energy expenditure in chow and high-fat-fed conditions at both 22 °C and 30 °C (thermoneutrality) and adjusted the energy expenditure data using multiple factors in our mouse models with loss of function mutations or genetic deletion of Mc3r (Lee et al., 2016). Results showed energy expenditure to be significantly decreased (p=0.048) only after adjusting for body weight at thermoneutrality and was not different in other conditions during chow feeding (Lee et al., 2016). In addition, our humanized MC3R^{hDM/hDM} mice exhibited no significant changes in total energy expenditure compared to controls and MC3R^{WT/hWT} in body weight, lean mass and lean + fat mass during high-fat feeding regardless of how the data accounted for body composition (Table 2b) (Lee et al., 2016). Therefore, we think that changes in energy expenditure play, at most, a minor role in the altered energy balance of MC3R^{hDM/hDM} mice (Table 2)(Lee et al., 2016). As we discussed here, we know that energy expenditure and the locomotion phenotype (Single Beam) was not robust in the *Mc3r^{TB/TB}* mice model. Thus, we did not anticipate that targeting the liver-specific role of *Mc3r* would have the potential to change in energy expenditure.*

*Our data confirmed previously reports that *Mc3r* deficient obesity mice model showed normal food intake. As described previously, energy expenditure and locomotor activity from both *Mc3r* deficient mice models, the global knockouts and the humanized knock-in *Mc3r* mouse model were analyzed from our independent studies in the introduction (Page 3-4, Line 51-58,60-69), results (Page 9, Line 198-206), and discussion section (Page 14-15, Line 337-348). All of these text changes are previously quoted in the response to Reviewer 1.*

3) The changes in fatty liver, insulin resistance and glucose levels are quite dramatic. Are these driven solely by the weight gain or is *Mc3r* affecting hepatic insulin sensitivity independent of body weight. Pair feeding studies would be one way to disentangle these effects.

Response: *Thanks for the reviewer comments. We now provide food intake monitoring and feeding efficiency (supplementary fig. 3c-d) data. We agree with the reviewer's concern. Most of the molecular and cellular analyses were performed in young 8-10 weeks mice before the weight gain, which allows us to understand the role of hepatic *Mc3r* prior to weight gain or obesity. We did not perform pair feeding studies from food intake data because intake has never been shown to be increased in the *Mc3r* deficiency models. We did not anticipate a change in pair feeding. To perform this experiment will take around 9 months to a year more. Instead, to demonstrate improvement in insulin sensitivity in *Mc3r* liver recovery mice, we added the data for glucose tolerance testing (GTT) (Fig. 3g-h) and free fatty acids (NEFA) (Fig. 4c). Furthermore, restoration of PGC-1 α and Cidec gene expression data (Fig. 4f) suggests that liver *Mc3r* recovery improves liver insulin sensitivity.*

4) Defective autophagy is suggested to drive the changes in hepatic steatosis and point to a role

for TFEB. The authors performed a series of in vivo and in vitro studies that do support this. However, without functional restoration of autophagy (potential by altering TFEB), it is difficult to determine if this is indeed the sole driver of the phenotype. Moreover, direct measurements of fatty acid esterification, oxidation and lipogenesis are necessary to further confirm that autophagy abnormalities are driving hepatic steatosis.

Response: We agree with the reviewer's comments. After TFEB translocation into the nucleus, it promotes the transcription of functional autophagy genes. Therefore, to test the function of TFEB by MC3R activation, we used gamma MSH-mediated TFEB translocation (Fig. 5j) and its gene transcription (Fig. 5l), without altering total TFEB expression levels (Supplementary Fig. 2b). We added the text in as a potential limitation to the discussion section (Page 15, Line 369-372).

“ Functional restoration of autophagy did not occur by modifying total TFEB expression. Further research is needed to determine whether TFEB is sole regulator responsible for mediating autophagy regulation in MC3R deficiency. In the future, more comprehensive research on MC3R-autophagy-TFEB regulation is warranted.”

Regarding direct measurements of fatty acid esterification, oxidation, and lipogenesis, we thank the reviewer for the wonderful suggestion. We have provided additional data for direct measurements of fatty acid esterification (CD36, Cidec), oxidation (PGC-1 α), Lipolysis (HSL) and lipogenesis (FASN) gene expression in the liver to confirm that autophagy abnormalities are driving hepatic steatosis (Fig. 4a-b) and systemic adiposity. We also confirmed serum NEFA (Fig. 4d) and glycerol levels (Fig. 4e) in circulation. Regulation of hepatic fatty acid metabolism and lipid droplet autophagy could explain such a significant systemic adiposity phenotype recovery. We added the text in the results (Page 9-10, Line 218-238; Page 9, Line 207-217) and discussion section (Page 14-15, Line 336-348).

Results

“We also examined hepatic mRNA expression of fatty acid metabolic genes (Figure 4f). CD36, the primary transporter responsible for the fatty acid esterification process, was significantly increased in Mc3r^{TB/TB} mice in the fasted state, though not in the fed state. Mc3r^{Hep/Hep} hepatic recovery mice showed significantly reduced CD36 in the fasted state compared to Mc3r^{TB/TB} with values similar to control mice (Figure 4f). These findings suggest that Mc3r deficiency leads to an overall increase in total fatty acid esterification. Liver lipid droplet surface binding protein Cidec (cell death inducing DFFA like effector C), was found to be upregulated, consistent with the excessive lipid droplet formation in Mc3r^{TB/TB}, as described previously (Begrache et al., 2011). Mc3r^{Hep/Hep} significantly recovered Cidec levels compared to Mc3r^{TB/TB}, which also suggests improvement in insulin sensitivity. In both wild-type control and liver recovery groups, the liver showed increased Cidec expression levels when mice were fasted. At the same time, Mc3r global deficiency did not respond to fasting and exhibited lower Cidec expression compared to the fed state, suggesting a defect in lipid droplet formation and lipid droplet autophagy. We observed no significant differences between Mc3r^{TB/TB} and controls for hepatic expression of the key rate-limiting enzymes of lipogenesis (fatty acid synthase; FASN) and of Lipolysis (HSL). In the fasted state, hepatic PGC-1 α was notably lower in

Mc3r^{TB/TB} than control mice but returned to a normal level in the hepatic recovery mice. This suggests that Mc3r deficiency may inhibit complete oxidation of fatty acids, which in turn affects the breakdown of excess fatty acids. The partial reversal of a significant systemic adiposity phenotype could potentially be explained by the regulation of hepatic fatty acid metabolism and lipid droplet autophagy.”

“To examine fat accumulation and tissue histology in liver specific Mc3r recovery, we evaluated hepatic TG accumulation relative to that of intact mice. Tissue TG content, markedly increased in Mc3r^{TB/TB} showed normal hepatic TG in Mc3r^{Hep/Hep}, suggesting restored hepatic lipid homeostasis (Fig. 4c). We also performed liver H&E staining (Fig. 4a) and Oil Red O staining (Fig. 4b); both showed elevated hepatic fat deposition in Mc3r^{TB/TB} that was reversed in the Mc3r^{Hep/Hep} mice. We also found that hepatic Mc3r recovery partially reversed the increased epididymal white adipose tissue (eWAT) of Mc3r^{TB/TB} (Supplementary Fig. 3e). Serum non-esterified fatty acid (NEFA) concentrations were significantly higher in Mc3r deficient mice; hepatic Mc3r recovery similarly normalized NEFA (Fig. 4d) without a concurrent change in serum glycerol concentrations (Fig. 4e) suggesting improving insulin sensitivity and hepatic fatty acid metabolism.”

Discussion

“The restoration of hepatic Mc3r resulted in improvements in fat mass, liver steatosis, fatty acid metabolism, and insulin sensitivity (as shown in Figures 3 and 4). However, we saw that the recovery of liver specific Mc3r did not affect food intake and feeding efficiency (as indicated in Supplementary Fig 3). Since, total daily energy expenditure is not significantly altered in MC3R-deficient mice (Butler et al., 2000; Sutton et al., 2006; Zhang et al., 2005), we did not anticipate there would be a hepatic-specific role of Mc3r in regulating energy expenditure. Recently, it was found that in the lateral hypothalamic area (LHA) and ventromedial hypothalamic area (VMH) Mc3r play crucial roles in regulating feeding, rheostasis, body weight maintenance, and metabolism (Dunigan et al., 2021; Pei et al., 2019). Energy expenditure was lower in Mc3r^{LHA} ablation mice without a change in food intake or fast–re-feeding food intake and accrual of body fat mass. These recent results confirm that Mc3r in major hypothalamic Mc3r-neurons may impact energy expenditure and locomotion (Dunigan et al., 2021; Pei et al., 2019). Thus, further investigation is required to elucidate if changes in energy expenditure and physical activity contribute to the phenotypic improvements noted in hepatic recovery mice.”

We hope with these changes our manuscript can now be considered acceptable for publication.

References:

Begrache, K., Levasseur, P.R., Zhang, J., Rossi, J., Skorupa, D., Solt, L.A., Young, B., Burris, T.P., Marks, D.L., Mynatt, R.L., and Butler, A.A. (2011). Genetic dissection of the functions of the melanocortin-3 receptor, a seven-transmembrane G-protein-coupled receptor, suggests roles for central and peripheral receptors in energy homeostasis. *J Biol Chem* 286, 40771-40781. 10.1074/jbc.M111.278374.

- Butler, A.A., Kesterson, R.A., Khong, K., Cullen, M.J., Pellemounter, M.A., Dekoning, J., Baetscher, M., and Cone, R.D. (2000). A unique metabolic syndrome causes obesity in the melanocortin-3 receptor-deficient mouse. *Endocrinology* *141*, 3518-3521. 10.1210/endo.141.9.7791.
- Collet, T.H., Dubern, B., Mokrosinski, J., Connors, H., Keogh, J.M., Mendes de Oliveira, E., Henning, E., Poitou-Bernert, C., Oppert, J.M., Tounian, P., et al. (2017). Evaluation of a melanocortin-4 receptor (MC4R) agonist (Setmelanotide) in MC4R deficiency. *Mol Metab* *6*, 1321-1329. 10.1016/j.molmet.2017.06.015.
- Dunigan, A.I., Swanson, A.M., Olson, D.P., and Roseberry, A.G. (2021). Whole-brain efferent and afferent connectivity of mouse ventral tegmental area melanocortin-3 receptor neurons. *J Comp Neurol* *529*, 1157-1183. 10.1002/cne.25013.
- Franco-Juarez, B., Coronel-Cruz, C., Hernandez-Ochoa, B., Gomez-Manzo, S., Cardenas-Rodriguez, N., Arreguin-Espinosa, R., Bandala, C., Canseco-Avila, L.M., and Ortega-Cuellar, D. (2022). TFEB; Beyond Its Role as an Autophagy and Lysosomes Regulator. *Cells* *11*. 10.3390/cells11193153.
- Ghamari-Langroudi, M., Cakir, I., Lippert, R.N., Sweeney, P., Litt, M.J., Ellacott, K.L.J., and Cone, R.D. (2018). Regulation of energy rheostasis by the melanocortin-3 receptor. *Sci Adv* *4*, eaat0866. 10.1126/sciadv.aat0866.
- Han, E.S., Evans, T.R., and Nelson, J.F. (1998). Adrenocortical responsiveness to adrenocorticotrophic hormone is enhanced in chronically food-restricted rats. *J Nutr* *128*, 1415-1420. 10.1093/jn/128.9.1415.
- Hatoum, I.J., Stylopoulos, N., Vanhoose, A.M., Boyd, K.L., Yin, D.P., Ellacott, K.L., Ma, L.L., Blaszczyk, K., Keogh, J.M., Cone, R.D., et al. (2012). Melanocortin-4 receptor signaling is required for weight loss after gastric bypass surgery. *J Clin Endocrinol Metab* *97*, E1023-1031. 10.1210/jc.2011-3432.
- Lee, B., Koo, J., Yun Jun, J., Gavrilova, O., Lee, Y., Seo, A.Y., Taylor-Douglas, D.C., Adler-Wailes, D.C., Chen, F., Gardner, R., et al. (2016). A mouse model for a partially inactive obesity-associated human MC3R variant. *Nat Commun* *7*, 10522. 10.1038/ncomms10522.
- Lee, M., Kim, A., Conwell, I.M., Hruby, V., Mayorov, A., Cai, M., and Wardlaw, S.L. (2008). Effects of selective modulation of the central melanocortin-3-receptor on food intake and hypothalamic POMC expression. *Peptides* *29*, 440-447. 10.1016/j.peptides.2007.11.005.
- Liu, K., and Czaja, M.J. (2013). Regulation of lipid stores and metabolism by lipophagy. *Cell Death and Differentiation* *20*, 3-11. 10.1038/cdd.2012.63.
- Marks, D.L., Hruby, V., Brookhart, G., and Cone, R.D. (2006). The regulation of food intake by selective stimulation of the type 3 melanocortin receptor (MC3R). *Peptides* *27*, 259-264. 10.1016/j.peptides.2005.01.025.

- Pei, H., Patterson, C.M., Sutton, A.K., Burnett, K.H., Myers, M.G., Jr., and Olson, D.P. (2019). Lateral Hypothalamic Mc3R-Expressing Neurons Modulate Locomotor Activity, Energy Expenditure, and Adiposity in Male Mice. *Endocrinology* *160*, 343-358. 10.1210/en.2018-00747.
- Renquist, B.J., Murphy, J.G., Larson, E.A., Olsen, D., Klein, R.F., Ellacott, K.L., and Cone, R.D. (2012). Melanocortin-3 receptor regulates the normal fasting response. *Proc Natl Acad Sci U S A* *109*, E1489-1498. 10.1073/pnas.1201994109.
- Singh, R. (2010). Autophagy and Regulation of Lipid Metabolism. In *Sensory and Metabolic Control of Energy Balance*, W. Meyerhof, U. Beisiegel, and H.G. Joost, eds. (Springer-Verlag Berlin), pp. 35-46. 10.1007/978-3-642-14426-4_4.
- Singh, R., Kaushik, S., Wang, Y., Xiang, Y., Novak, I., Komatsu, M., Tanaka, K., Cuervo, A.M., and Czaja, M.J. (2009). Autophagy regulates lipid metabolism. *Nature* *458*, 1131-1135. 10.1038/nature07976.
- Sutton, G.M., Trevaskis, J.L., Hulver, M.W., McMillan, R.P., Markward, N.J., Babin, M.J., Meyer, E.A., and Butler, A.A. (2006). Diet-genotype interactions in the development of the obese, insulin-resistant phenotype of C57BL/6J mice lacking melanocortin-3 or -4 receptors. *Endocrinology* *147*, 2183-2196. 10.1210/en.2005-1209.
- Sweeney, P., Bedenbaugh, M.N., Maldonado, J., Pan, P., Fowler, K., Williams, S.Y., Gimenez, L.E., Ghamari-Langroudi, M., Downing, G., Gui, Y., et al. (2021). The melanocortin-3 receptor is a pharmacological target for the regulation of anorexia. *Sci Transl Med* *13*. 10.1126/scitranslmed.abd6434.
- Zhang, Y., Kilroy, G.E., Henagan, T.M., Prpic-Uhing, V., Richards, W.G., Bannon, A.W., Mynatt, R.L., and Gettys, T.W. (2005). Targeted deletion of melanocortin receptor subtypes 3 and 4, but not CART, alters nutrient partitioning and compromises behavioral and metabolic responses to leptin. *FASEB J* *19*, 1482-1491. 10.1096/fj.05-3851com.

We thank you for the opportunity to respond to the reviewers' comments. We have revised the paper according to their helpful suggestions as follows:

Reviewer #1 (Remarks to the Author):

In the revised version of manuscript, the authors performed additional experiments to address concerns raised by this and the other reviewer. The manuscript has been improved but there are still several unresolved issues as specified below.

1. It is still unclear how obesity observed in Mc3r-deficient mice are reversed by the re-expression of Mc3r in the liver only. As noted by the authors in Figs. 1a and 1b, expression level of Mc3r in the liver is quite low. While this level of Mc3r may have a role in the liver, as the authors demonstrated, it is not very likely that this level of Mc3r can affect body weight and whole-body adiposity when Mc3r is absent everywhere else. The authors showed additional data in this revision, but the most important data – changes in energy expenditure – to explain the observed phenotype is missing. One may readily understand that re-expression of Mc3r in the liver rescues adiposity and autophagy of the liver, but if the authors want to claim that body weight and whole-body adiposity are reversed by the re-expression of Mc3r in the liver, the mechanism for this change is key to support it. Therefore, it is necessary to examine what happens to the energy expenditure (for example, as determined by indirect calorimetry) and locomotion. It may also be a good idea to look at BAT.

Response: *As per reviewer suggestion, we have now provided a detailed assessment of energy expenditure, energy intake, and locomotor activity at both ambient 22 °C and thermoneutral temperature 30 °C by indirect calorimetry (Figure 4 and Supplementary Fig. 4a-f). Energy expenditure and energy intake data were adjusted for body weight, lean and fat mass, and lean mass (Supplementary Table 1). Additionally, we have monitored brown adipose tissue (BAT) cold-induced UCP-1 activation by immunofluorescence and immunoblotting (Supplementary Fig. 4g-i). We have added the text describing the new results section in the introduction, results, methodology, and discussion sections is highlighted in grey and as follows:*

Introduction:

Previous studies have demonstrated that Mc3r global knockout mice do not have excessive food intake; rather they are described as being hypophagic relative to controls^{13, 15, 23}. Additionally, some studies show evidence for reduced energy expenditure, respiratory exchange ratio, and locomotion in Mc3r^{TB/TB} mice²³. However, other studies have demonstrated normophagia and normal global metabolism in Mc3r deficiency²⁴, leaving the mechanism of commonly observed increased adiposity undetermined^{14, 23, 25, 26}.

Results

Effect of MC3R deficiency in regulating feeding behavior

We monitored food intake for 15 days in single-housed mice, which was not significantly different in Mc3r^{TB/TB} and Mc3r^{HEP/HEP} mice compared to Mc3r^{+/+} (Supplementary Fig. 3c), even after adjusting for body weight or lean mass. Additionally, feeding efficiency (FE) did not differ between groups (Supplementary Fig. 3d). However, when food intake monitoring was conducted using CLAMS metabolic chambers (Fig. 4a-c), food intake differences emerged. Daily energy

intake was significantly reduced in $Mc3r^{TB/TB}$ and $Mc3r^{HEP/HEP}$ mice compared to $Mc3r^{+/+}$ at 22 °C (Fig. 4b). After adjusting energy intake for body weight, lean & fat mass, or lean mass, $Mc3r^{TB/TB}$ and $Mc3r^{HEP/HEP}$ remained significantly lower than $Mc3r^{+/+}$ (Supplementary Table 1). At thermoneutrality (30 °C), energy intake was significantly reduced in $Mc3r^{TB/TB}$ vs. $Mc3r^{+/+}$ only during the dark phase (Fig. 4c). This reduction in energy intake is consistent with previously reported findings that global transcriptionally blocked $Mc3r$ -deficient mice are relatively hypophagic^{23, 25}. Here, re-expression of $Mc3r$ only in the liver did not restore energy intake, suggesting that feeding behavior alterations in $Mc3r$ deficiency are centrally (CNS) regulated and not via hepatic regulation²³.

Hepatic MC3R and energy expenditure, locomotion, and brown adipose tissue thermogenesis
 To examine whether differences in total energy expenditure (TEE) contribute to the altered energy balance seen in $Mc3r^{HEP/HEP}$, we performed indirect calorimetry at ambient 22 °C and thermoneutral temperature 30 °C during chow feeding using $Mc3r^{+/+}$, $Mc3r^{TB/TB}$, and $Mc3r^{HEP/HEP}$ mice (Fig. 4d). TEE was significantly reduced in $Mc3r^{TB/TB}$ mice in the dark phase, which also affected total (24-hour EE) at ambient 22 °C and thermoneutral temperature 30 °C (Fig. 4e-f). This TEE difference persisted at thermoneutral condition (30°C) after adjusting for body weight and total lean & fat mass but not for lean mass (Supplementary Table 1). However, for $Mc3r^{HEP/HEP}$ mice, total energy expenditure was intermediate, and therefore not significantly different compared to either $Mc3r^{+/+}$ or $Mc3r^{TB/TB}$ groups after adjusting for body weight, lean & fat mass, or lean mass at both temperature conditions (Supplementary Table 1). Total oxygen consumption (VO_2) was significantly reduced in $Mc3r^{TB/TB}$ mice at ambient 22 °C and thermoneutral temperature 30 °C (Fig. 4g-i). After adjusting VO_2 for body weight or total lean & fat mass, $Mc3r^{TB/TB}$ significantly differed from $Mc3r^{+/+}$ at 22 °C or 30 °C (Supplementary Table 1). VO_2 was intermediate, and therefore not significantly different in $Mc3r^{HEP/HEP}$ mice in the light and dark phase, and for total daily VO_2 (Fig. 4g-i), even after adjusting for body weight, lean & fat mass, and lean mass at both temperature conditions (Supplementary Table 1). Respiratory exchange ratio (RER) appeared to show a trend toward lower total RER in $Mc3r^{TB/TB}$ compared to $Mc3r^{+/+}$, while $Mc3r^{HEP/HEP}$ mice were significantly increased compared to $Mc3r^{TB/TB}$ mice at 30 °C and not at 22 °C primarily due to increased RER of $Mc3r^{HEP/HEP}$ in the dark phase (Fig. 4j-l). Restored RER in $Mc3r^{HEP/HEP}$ suggests that altered substrate preference (from fat to carbohydrates) for generating energy appears to be a consistently applicable mechanism underlying the altered energy balance of $Mc3r^{HEP/HEP}$ as compared to $Mc3r^{TB/TB}$ mice. We also monitored locomotion activity by beam breaks in the metabolic chambers during the light and dark phase cycles at 22 °C and 30 °C using $Mc3r^{+/+}$, $Mc3r^{TB/TB}$, and $Mc3r^{HEP/HEP}$ mice. Total activity and ambulatory activity (physical activity) were all similar between groups during both light and dark phases (Supplementary Fig. 4a-f), suggesting that locomotion is not a significant factor in the change in the energy homeostasis regulation and obesity phenotype of $Mc3r$ deficiency. Lastly, we looked at brown adipose tissue (BAT) UCP-1 activity to monitor thermogenesis after studying mice either at ambient 22 °C or after cold exposure (6 °C) for 6 hours. UCP-1 staining increased after cold exposure compared to ambient temperature in all groups without notable differences in the signal intensity between $Mc3r^{+/+}$, $Mc3r^{TB/TB}$, and $Mc3r^{HEP/HEP}$ in either temperature condition (Supplementary Fig. 4g). Immunoblotting results for UCP-1 protein expression were also similar between the groups (Supplementary Fig. 4h-i). $Mc3r$ deficiency did not significantly alter BAT UCP-1 activity, one of the potential mechanisms for dissipating energy as heat.

Discussion:

Recently, it was found that *Mc3r* expression in the lateral hypothalamic area (LHA) and ventromedial hypothalamic area (VMH) play crucial roles in regulating feeding, rheostasis, body weight maintenance, and metabolism^{23, 25, 51, 52}. Our data confirmed relative hypophagia in transcriptionally blocked global *Mc3r* deficient mice studied in metabolic chambers (using real-time monitoring feeder hangers)²³. Surprisingly, we did not observe these changes in food intake in singly-housed mice in our colony during a two-week observation period, using weighed food pellets. We found that the recovery of liver-specific *Mc3r* did not reverse lowered energy intake (as indicated in Fig 4a-c), and thus, hepatic re-expression of *Mc3r* did not restore feeding behavior, supporting the previous finding that *Mc3r* deficiency is related to appetite regulation by major hypothalamic MC3R neurons²³. We found that total energy expenditure and oxygen consumption were reduced in global *Mc3r* deficient mice even after adjusting for body weight and total lean plus fat mass. Total energy expenditure measured using indirect calorimetry was lower in *Mc3r^{TB/TB}* in the mid-dark period and even after adjustment for body weight²³. From the above observations, daily energy intake matches the 24-hour total energy expenditure measured (if calculated daily), suggesting feeding behavior and energy expenditure are similar phenotypes between global knockouts and hepatic recovery mice. However, it remains possible that the small differences in energy expenditures as observed here might contribute to the more-normal body composition of *Mc3r^{Hep/Hep}*. Although previous works have shown that locomotion may be reduced in *Mc3r^{TB/TB}* mice during the dark period²³, we did not see a change in energy expenditure, oxygen consumption, and locomotion activity after adjusting for body composition in our hepatic recovery mouse model; thus, hepatic *Mc3r* reactivation is not able to reverse the whole-body energy metabolism differences of *Mc3r* deficiency entirely. This suggests that other extrahepatic peripheral tissues and central MC3R neurons are important areas of interest for MC3R-mediated energy homeostasis and metabolism. Furthermore, energy expenditure was lower in *Mc3r^{LHA}* ablation mice without a change in food intake or fast-refeeding food intake and accrual of body fat mass. Our results are thus consistent with prior observations that *Mc3r* in major hypothalamic *Mc3r*-neurons may impact energy expenditure^{51, 52}. RER is an indicator of oxidizing fatty acids and trended lower in *Mc3r^{TB/TB}*. The increased RER in *Mc3r^{HEP/HEP}* suggests a shift in substance preference. Hepatic reactivation of *Mc3r* restored lipid recycling and shifted metabolism towards using carbohydrates for fuel.

Methods:

Energy expenditure, energy intake, and locomotion activity: Total energy expenditure (TEE), energy intake (calculated using metabolizable energy of diet 3.05 kcal/g), oxygen consumption rate (VO₂), respiratory exchange ratio (RER), and physical activity (infrared beam breaks as total activity, 1-inch spacing) were measured with an indirect calorimetry system (CLAMS-HC Oxyman v5.52 software, Columbus Instruments) as previously described⁵⁷. Prior to the metabolic measurements, mice were acclimated in the metabolic chambers at room temperature for 2 days, so the data recorded then were excluded from daily energy expenditure final calculations. On day 3, metabolic parameters were recorded (260 s intervals) continuously at 22°C for 24 hours. On day 4, the chamber temperature was elevated to 30°C at 6 am and recorded for additional 24 h (the first hour after temperature change was excluded from analysis) Mice were housed individually with ad libitum access to food (hanging feeder) and water in Tecniplast 1284 cages with ~95 g of wood chip bedding (7090 Teklad sani-chips, Envigo, Indianapolis, IN) with measured physical activity (infrared beam breaks as total activity, 1 inch spacing) and continual monitoring of total activity in each cage. Calorimetry parameters are: 7.75 L volume, 0.9 L/min flow rate, 0.6 L/min sampling flow, 15 s settle time, 5 s measure time, with each chamber sampled every 260 s. Thus, the physical activity was measured per 260 s interval, giving 14 sampling cycles per 61 min interval. All 12 calorimetry chambers were housed in a single temperature-controlled environmental chamber.

Brown adipose tissue (BAT) immunofluorescence procedure: Standard protocols for immunofluorescence of BAT were followed with some adaptations. Mice were housed at either 22°C or exposed to 6°C for 6 hours. After temperature exposure, mice were perfused using 1x PBS followed by 4% paraformaldehyde (PFA); then, BAT samples were fixed overnight in 4 % PFA at 4°C. Cryoprotection of BAT was achieved by immersion in 10%, 20%, and 30% (w/v) sucrose overnight, followed by embedding in OCT and freezing using Tissue-Tek Cryo 3 Flex Cryostat for sectioning. For UCP-1 expression detection, 10 µm frozen sections were airdried for 30 minutes at room temperature. Sections were blocked with 5% horse serum and incubated in a humidified chamber with UCP-1 (1:250) and perilipin (1:500) antibodies overnight at 4°C (Supplementary Table 2). Following washing steps, anti-rabbit Alexa Fluor 488 (Thermo Fisher Scientific #A1100) and mouse Alexa Fluor 647 (Thermo Fisher Scientific # A21236) secondary antibodies (1:500) were incubated for 30 mins. Following washing steps, slides were treated with DAPI solution (300 nM in 0.1% TBS-T) for 3 minutes at room temperature to visualize cell nuclei and mounted with VectoMount express mounting medium (Vector Laboratories, Inc). Imaging was performed using a fluorescence microscope at 20X magnification.

Statistical analysis: Energy expenditure data was examined using analysis of covariance (ANCOVA) adjusted for body weight, lean & fat mass, and lean mass, and post hoc tests for estimated marginal means were calculated using SPSS 28.0.2.

2. Supplemental Figure 3d, the authors stated that daily food intake was not changed in the Mc3r-deficient mice, but that feeding efficiency tended to be increased when both are not significantly different between control and Mc3r-deficient mice (page 9, lines 202-206). Please revise this part of the manuscript to accurately describe the findings.

Response: We revised this part of the manuscript to accurately describe the findings (page 9, lines 204-205) and incorporated the new findings from metabolic chambers data:

We monitored food intake for 15 days in single-housed mice, which was not significantly different in Mc3r^{TB/TB} and Mc3r^{HEP/HEP} mice compared to Mc3r^{+/+} (Supplementary Fig. 3c), even after adjusting for body weight or lean mass. Additionally, feeding efficiency (FE) did not differ between groups (Supplementary Fig. 3d). However, when food intake monitoring was conducted using CLAMS metabolic chambers (Fig. 4a-c), food intake differences emerged. Daily energy intake was significantly reduced in Mc3r^{TB/TB} and Mc3r^{HEP/HEP} mice compared to Mc3r^{+/+} at 22 °C (Fig. 4b). After adjusting energy intake for body weight, lean & fat mass, or lean mass, Mc3r^{TB/TB} and Mc3r^{HEP/HEP} remained significantly lower than Mc3r^{+/+} (Supplementary Table 1). At thermoneutrality (30 °C), energy intake was significantly reduced in Mc3r^{TB/TB} vs. Mc3r^{+/+} only during the dark phase (Fig. 4c). This reduction in energy intake is consistent with previously reported findings that global transcriptionally blocked Mc3r-deficient mice are relatively hypophagic^{23, 25}. Here, re-expression of Mc3r only in the liver did not restore energy intake, suggesting that feeding behavior alterations in Mc3r deficiency are centrally (CNS) regulated and not via hepatic regulation²³.

3. Figure 1a, Mc3r expression levels are not significantly different at ZT12 and ZT2. Therefore, the authors' interpretation that Mc3r expression has diurnal variation (page 5, lines 95-99) is not correct unless they perform more experiments to obtain statistical significance.

Response: We corrected and removed the “diurnal variation in the Mc3r expression” from the text (page 5, lines 93-98).

Although the estimated total Mc3r transcript levels were relatively lower in liver compared to hypothalamus (Fig. 1b), it was significantly higher in livers from control mice compared to those of Mc3r^{TB/TB} mouse livers at Zeitgeber time (ZT) 12 and not at ZT2 (Fig. 1a). Notably, we found that expression of Mc3r transcripts in the control liver (Fig. 1a) was high at nighttime (ZT12), when MC3R ligands circulate in rodent plasma^{32, 33}.

4. Figures 1g-1h and Supplemental Figures 1a-1b, why do Mc3r-deficient livers have higher numbers of lipid droplet numbers in Figures 1g-1h, but not in Supplemental Figures 1a-1b? Were the experiments in Figures 1g-1h performed in fed conditions? If so, it should be mentioned in the text and/or figure legends.

Response: Thank you for this question. We now make clear that the experiments in Figures 1g-1h were performed in fed conditions in the text and figure legends.

Lines 107-109: *The elevated liver TG of Mc3r insufficiency was associated with increased LD deposition, as detected by transmission electron microscopy (TEM) in liver cells in the fed condition (Figs 1g-h).*

Reviewer #2 (Remarks to the Author):

Thank you to the authors for the additional data and textual clarifications to improve the manuscript. One significant concern remains unaddressed. As also noted by reviewer 1, the ability of re-expression of Mc3r to reverse the obesity phenotype is very surprising (and a main crux of the paper). This phenotype should be supported by complementary analysis of energy balance. It is appreciated that additional cohorts of mice are required that will take several months to generate but these are important physiological studies that (in my opinion) are necessary to support the conclusions of the main paper.

Response: *In response to reviewers' comments, we have now provided a detailed assessment of energy expenditure, energy intake, and locomotor activity at both ambient 22 °C and thermoneutral temperature 30 °C by indirect calorimetry (Figure 4 and Supplementary Fig. 4a-f). Energy expenditure and energy intake data were adjusted for body weight, lean and fat mass, and lean mass (Supplementary Table 1). We have added the text describing the new results section in the introduction, results, methodology, and discussion sections as indicated in the response to Reviewer 1.*

Additional changes:

We have also made some additional changes to the text for clarity and to add information, for instance adding that, "...recent studies using thermal proteome profiling revealed that the top reactome enriched after MC3R activation is autophagy proteins ⁴⁸."

We hope that with these additional work and revised manuscript, our manuscript can now be considered acceptable for publication.

We thank you for the opportunity to respond to the reviewers' and editor's comments. We have revised the paper according to these helpful suggestions as follows:

Reviewer #1 (Remarks to the Author):

1. The authors performed additional experiments to assess energy expenditure, locomotor activity, and BAT phenotypes in this round of revision. In the new figure 4, the authors reported small but significant decreases of energy intake, total energy expenditure, and oxygen consumption in the Mc3r^{TB/TB} mice compared to Mc3r^{+/+} mice. These phenotypes were not reversed in Mc3r^{Hep/Hep} mice. Therefore, it still remains unanswered how re-expression of MC3R only in the liver (Mc3r^{Hep/Hep} mice) reverses body weight as reported in figures 3i and 3j. The authors claim in discussion (page 16, lines 401-403) that “However, it remains possible that the small differences in energy expenditures as observed here might contribute to the more-normal body composition of Mc3r^{Hep/Hep}.”, but it is not clear if this small and insignificant differences of energy expenditure may result in such a robust restoration phenotype of body weight.

Response: We thank the reviewer for helping strengthen the study and encouraging us to provide the necessary studies that explain how liver Mc3r reactivation affects whole body energy balance. We first added more mice numbers per genotype to our indirect calorimetry experiments, which resulted in statistical significance between groups, where previously we could show only trends observed in Mc3r^{Hep/Hep} mice. We can now demonstrate significant findings, including significant improvements towards normal for Mc3r^{Hep/Hep} mice in total energy expenditure, oxygen consumption, and respiratory exchange ratio. We have also added a study that demonstrates improvement in adaptive energy expenditure regulation in Mc3r^{Hep/Hep} mice after mice were cold-exposed. These findings support the conclusion that Mc3r is key in adapting to energy demanding conditions. Moreover, we provide additional evidence showing normalized tissue weights (liver and eWAT) and serum lipid profile of total cholesterol, triglycerides, and non-esterified free fatty acids in Mc3r^{Hep/Hep} mice.

To elaborate on the mechanism behind altered energy expenditure, we hypothesized that it may result from metabolic adaptations to changes in energy production. We studied mitochondrial respiration using the Seahorse Mito Stress Test to examine the basal and maximal respiration of cultured primary hepatocytes, which reflects the highest rate of oxygen consumption that a cell can achieve under stressed conditions. Mc3r-deficient mice exhibited reduced basal and maximal oxygen consumption in hepatocytes, suggesting altered mitochondrial function and energy production from mitochondrial respiration. Re-expression of the Mc3r protein in hepatocytes restored mitochondrial respiration and improved mitochondrial efficiency and capacity, highlighting the important role of the Mc3r protein in cellular energy metabolism. The recovery of mitochondrial respiration and capacity may indicate a restoration of overall mitochondrial function and, consequently, energy metabolism, leading to the efficient utilization of alternative sources such as fatty acids, glucose, and glutamine to meet ATP needs.

A liver RNAseq analysis has also been added, showing that an examination of differentially expressed genes confirms that Mc3r^{TB/TB} mice are different from both Mc3r^{+/+} and Mc3r^{Hep/Hep}

mice, while $Mc3r^{Hep/Hep}$ mice are similar to $Mc3r^{+/+}$ mice. Notably, the gene ontology analysis highlights pathways related to lipid droplet autophagy, lipid metabolism, and energy homeostasis, showing that $Mc3r$ deficiency frequently differs from both $Mc3r$ reactivation and wild type in the same systems we identified by our prior experiments. We added results from liver tissue DESeq2, showing that expression of TFEB downstream target genes related to autophagy and lysosomal-associated membrane protein expression genes are restored in the hepatic $Mc3r$ reactivation model. These data strengthen the argument that liver autophagy is a predominant mechanism for fat accumulation and fat partitioning in $Mc3r$ deficiency and is sufficient to explain the significant effects in obesity and metabolic syndrome we have observed.

We have added the text describing the new results section in the results and discussion sections as follows:

Results

Effect of hepatic $Mc3r$ reactivation on energy expenditure, locomotion, and brown adipose tissue thermogenesis.

To examine whether differences in total energy expenditure (TEE) contribute to the altered energy balance seen in $Mc3r^{HEP/HEP}$, we performed indirect calorimetry at ambient 22 °C and thermoneutral temperature 30 °C during chow feeding using $Mc3r^{+/+}$, $Mc3r^{TB/TB}$, and $Mc3r^{Hep/Hep}$ mice (Fig. 5d). TEE was significantly reduced in $Mc3r^{TB/TB}$ mice compared to $Mc3r^{+/+}$ in both light and dark phases. Notably, with $Mc3r$ reactivation, $Mc3r^{Hep/Hep}$ mice exhibited a significant increase in TEE compared to $Mc3r^{TB/TB}$ in both light and dark phases, which also translated to total 24-hour energy expenditure (EE) at ambient 22 °C (Fig. 5e-f). For TEE at a thermoneutral temperature of 30 °C, $Mc3r^{TB/TB}$ was significantly lower compared to both $Mc3r^{+/+}$ and $Mc3r^{Hep/Hep}$ groups in the light phase and for total 24-hour EE (Fig. 5e-f). Recovery from reduced 24-hour total energy expenditure in the $Mc3r^{TB/TB}$ phenotype indicates that the role of hepatic $Mc3r$ in autophagy is a key mechanism involves in lipid metabolism, energy expenditure and body weight. For $Mc3r^{TB/TB}$ mice, TEE was also reduced significantly compared to both $Mc3r^{+/+}$ or $Mc3r^{Hep/Hep}$ groups after adjusting for body weight and lean & fat mass at both temperature conditions (Supplementary Table 1). At a thermoneutral temperature of 30°C, total energy expenditure (TEE), adjusted for body weight and lean and fat mass, showed a trend for an intermediate phenotype in $Mc3r^{Hep/Hep}$; however, it was not statistically significant compared to $Mc3r^{TB/TB}$.

Compared to control mice (Fig. 5g-i), total oxygen consumption (VO_2) was significantly reduced in $Mc3r^{TB/TB}$ mice, but not in $Mc3r^{Hep/Hep}$ mice, at ambient 22 °C and 30 °C. Total VO_2 was significantly increased in $Mc3r^{Hep/Hep}$ versus $Mc3r^{TB/TB}$ mice at ambient 22 °C (Fig. 5g-i). After adjusting VO_2 for body weight, and total lean & fat mass, $Mc3r^{TB/TB}$ also significantly differed from $Mc3r^{+/+}$ at both temperatures. At a thermoneutral temperature of 30°C, $Mc3r^{Hep/Hep}$ mice's total VO_2 exhibited a significant recovery relative to $Mc3r^{TB/TB}$, even after adjustment for body weight and total lean & fat mass, confirming the significant recovery phenotype (Supplementary Table 1). This difference in VO_2 consumption is a key indicator of improvement in metabolic activity. Respiratory exchange ratio (RER) appeared to show lower total RER in $Mc3r^{TB/TB}$ compared to $Mc3r^{+/+}$, while $Mc3r^{Hep/Hep}$ mice were significantly increased compared to $Mc3r^{TB/TB}$ mice at 22 °C in the dark phase and 24-hour RER at 30 °C (Fig. 5j-l). Restored RER in $Mc3r^{Hep/Hep}$ implies an altered substrate preference (from fat to carbohydrates) for generating energy, as might be expected in a mouse with less adipose tissue.

Hepatic $Mc3r$ reactivation restores adaptive energy expenditure, cellular respiration, and lipid droplet autophagy.

To further investigate the critical role of hepatic MC3R in energy balance, we studied how *Mc3r* deficiency affects adaptive energy expenditure in response to cold exposure, when the body requires energy for thermogenesis. Adaptive energy expenditure (EE) in response to cold exposure was significantly blunted in *Mc3r^{TB/TB}* compared to *Mc3r^{+/+}* mice, even within the first few hours (Fig 6a-b). In contrast, *Mc3r^{Hep/Hep}* mice normally responded to cold exposure EE compared to *Mc3r* deficient mice after 6 hours at 6°C (Fig. 6a-b). Corroborative results were found from oxygen consumption measurements. Oxygen consumption (VO_2) was significantly reduced in *Mc3r^{TB/TB}* mice and significantly different from *Mc3r^{+/+}* and *Mc3r^{Hep/Hep}* at 6 °C (Fig. 6c-d). These results suggest that hepatic *Mc3r* is essential for regulating adaptive energy expenditure in response to thermogenic energy demands.

To elaborate the mechanism for altered energy expenditure at the hepatocyte level, we cultured primary hepatocytes to study cellular mitochondrial respiration using the Seahorse Mito Stress Test. Compared to *Mc3r^{+/+}* and *Mc3r^{Hep/Hep}* group, *Mc3r^{TB/TB}* cells showed reduced basal level respiration (Fig. 6e). We observed restoration in basal respiration, measured by the oxygen consumption rate, in *Mc3r^{Hep/Hep}* hepatocytes compared to the *Mc3r^{TB/TB}* group (Fig. 6f). Injection of the mitochondrial uncoupler FCCP allows a measurement of maximum respiration which reflects mitochondrial ability to respond to increased energy demands (Fig. 6e). Maximum respiration was also restored to normal in *Mc3r^{Hep/Hep}*, which was significantly greater than *Mc3r^{TB/TB}* group and similar to the *Mc3r^{+/+}* control group (Fig. 6g). These Seahorse assay results in hepatocytes align with the energy expenditure data observed in the mouse models, further validating the importance of liver *Mc3r* for energy regulation. These findings strongly support the conclusion that hepatic *Mc3r* plays a crucial role in adapting to external stimuli, contributing significantly to systemic obesity through its effects on peripheral tissues.

To understand the role of *Mc3r* using an unbiased approach, bulk RNA-sequencing analysis was performed on liver samples from *Mc3r^{+/+}*, *Mc3r^{TB/TB}*, and *Mc3r^{Hep/Hep}* genotypes. Log₂ fold change (Log₂FC) values for all significantly differentially expressed genes were plotted in a scatter plot. Linear regression fit line comparisons between *Mc3r^{TB/TB}* vs. *Mc3r^{+/+}* and *Mc3r^{Hep/Hep}* vs. *Mc3r^{+/+}* (blue line, $r = 0.189$, $P < 0.0001$) suggest that *Mc3r^{TB/TB}* and *Mc3r^{Hep/Hep}* are drastically different (Fig. 6h). Linear regression fit lines for the comparisons between *Mc3r^{TB/TB}* vs. *Mc3r^{+/+}* and *Mc3r^{Hep/Hep}* vs. *Mc3r^{TB/TB}* (orange line, $r = 0.726$, $P < 0.0001$) slightly overlapped with the best-fit perfect correlation line, indicate that *Mc3r^{+/+}* and *Mc3r^{Hep/Hep}* are largely similar (Fig. 6i). These data suggest that the hepatic changes in differentially expressed genes caused by *Mc3r* global knockout are largely reversed by the recovery of hepatic *Mc3r*.

Next, we performed gene ontology (GO) biological pathway analysis for the comparisons between *Mc3r^{TB/TB}* vs. *Mc3r^{+/+}*, *Mc3r^{Hep/Hep}* vs. *Mc3r^{+/+}* and *Mc3r^{Hep/Hep}* vs. *Mc3r^{TB/TB}*. A heatmap of negative adjusted log₁₀ *P* values shows that the *Mc3r^{Hep/Hep}* and *Mc3r^{+/+}* groups had greater similarity (Fig. 6j). In contrast, *Mc3r^{TB/TB}* shows much lower *p*-values that are significantly different compared to *Mc3r^{+/+}* and *Mc3r^{Hep/Hep}*, particularly in pathways related to lipid droplet autophagy and lipid metabolism (Fig. 6j). These findings reinforce the notion that liver autophagy is a predominant mechanism influencing fat accumulation and lipid partitioning in MC3R deficiency.

TFEB signaling affected by MC3R pathway in hepatic autophagy regulation

To further investigate precise candidate mechanisms underlying the hepatic *Mc3r* reactivation model, we performed DESeq2 analysis on bulk RNA-sequencing data from liver tissue. Notably, TFEB-driven autophagy-related genes, including *Mcoln1* (mucopolipin 1), *Neu1* (neuraminidases), and *Sqstm1* (p62 protein), showed increased expression in *Mc3r^{Hep/Hep}* mice compared to *Mc3r^{TB/TB}* mice (Fig 7l). Interestingly, lysosomal-associated membrane proteins,

Lamp1 and *Lamp2* showed expression was elevated in *Mc3r^{TB/TB}* mice compared to both *Mc3r^{+/+}* and *Mc3r^{Hep/Hep}* groups (Fig 7I).

Discussion:

Liver RNA-seq analysis revealed that differentially expressed genes (DEGs) were significantly altered in global knockout mice, while the DEG profile of hepatic reactivation mice was closer to that of the control group. Gene ontology (GO) pathway analysis suggests that the affected pathways are related to lipid droplet autophagy, macroautophagy, and lipid metabolism, indicating that *Mc3r* deficiency leads to distinct changes compared to either *Mc3r* reactivation or wild-type mice.

Total energy expenditure, oxygen consumption and RER were reduced in global *Mc3r* deficient mice even after adjusting for body weight and total lean plus fat mass. Total energy expenditure measured using indirect calorimetry was lower in *Mc3r^{TB/TB}* in the mid-dark period and even after adjustment for body weight²³. Hepatic *Mc3r* reactivation improved total energy expenditure, oxygen consumption, and RER. These differences in total energy expenditure, oxygen consumption, and RER, as observed here, may contribute to the less obese body composition of *Mc3r^{Hep/Hep}*. RER is an indicator of oxidizing fatty acids and is significantly lower in *Mc3r^{TB/TB}*. The increased RER in *Mc3r^{Hep/Hep}* suggests a shift in substance preference. Hepatic reactivation of *Mc3r* restored lipid recycling and shifted metabolism towards using carbohydrates for fuel. Improvement in energy expenditure-related parameters suggests a potential mechanism to improve body weight, fat mass, and systemic adiposity.

In a similar line, we assessed cellular respiration in primary hepatocytes. *Mc3r* deficient mice exhibited reduced oxygen consumption, suggesting a reduced metabolic demand and a decrease in their maximal electron transport capacity, as these cells are not fully utilizing their mitochondrial capacity under normal conditions. Notably, in red oxidative muscle isolated from gastrocnemius, *Mc3rKO* mice were previously shown to have a decrease in fatty acid oxidation and citrate synthase activity, consistent with reduced mitochondrial content¹⁴.

Primary hepatocytes isolated from mice exposed to a 60% high-fat diet (HFD) for 12 weeks (who generally develop diabetes) have been reported to have reduced basal respiration as well as reduced ATP-linked respiration (oxidative phosphorylation), and their response to the uncoupler FCCP was also decreased when mitochondrial energy metabolism assessed via extracellular flux analysis⁵⁶. However, mitochondria isolated from mice with liver steatosis from a 35% high-fat and high-sugar diet for 5 months (who mostly show insulin resistance without diabetes) have normal levels of mitochondrial respiratory chain complex I–V proteins and do not demonstrate adversely impacted mitochondrial respiration⁵⁷. These data suggest that hepatic mitochondria may initially activate compensatory mechanisms to counteract liver damage associated with severe hepatic steatosis. However, persistent fat accumulation can eventually surpass this adaptive capacity, leading to impaired mitochondrial function due to an overload of free fatty acids (FFA)⁵⁸. Indeed, primary human hepatocytes exposed *in vitro* to FFAs showed suppressed maximal respiration and maximum fatty acids beta-oxidation, further indicating compromised mitochondrial function⁵⁹. Consequently, reductions in liver mitochondrial State 3 respiration can be found⁶⁰, particularly in mice with high histological grade MAFLD⁶¹. However, there are also rodent models, e.g., the Otsuka Long-Evans Tokushima fatty rat⁶² and mice with heterozygous inactivation of the mitochondrial trifunctional protein⁶³, where mitochondrial dysfunction is present before insulin resistance and steatosis develops, and is believed to contribute etiologically to the development of MAFLD.

Dietary fats absorbed by the gut are transported to the liver for proper processing, packaging into lipoproteins, and circulation. In the liver, autophagy is the primary mechanism for the catabolism of lipoproteins, although lipolysis also occurs, though to a lesser extent. Reactivation of Mc3r leads to improved lipid droplet autophagy; here, we showed how Mc3r insufficiency affected cellular and molecular mechanisms of autophagy. Due to defective Mc3r-mediated lipid droplet autophagy, hepatocytes do not efficiently metabolize triglycerides. By restoring defective autophagy, we observed reductions in liver triglyceride content and liver weight and partial improvements in eWAT weight. Our results demonstrated the restoration of normal circulating non-esterified fatty acid concentrations, total cholesterol, and triglycerides in Mc3r^{Hep/Hep} mice. These suggest that Mc3r reactivation largely restores lipid metabolism in hepatocytes and leads to reduced excess circulating fats that would be stored by extrahepatic tissues (e.g., eWAT). The partial recovery of body weight, fat mass, and total energy expenditure at the systemic level underscores how hepatic Mc3r reactivation can modulate a systemic obesity phenotype.

We also showed that expression of TFEB downstream target genes for autophagy-related and lysosomal-associated membrane proteins is restored in hepatic *Mc3r* re-expression:

Moreover, the findings of MC3R-dependent nuclear TFEB allocation (Fig. 7j) with subsequent autophagy and lysosomal gene activations (Fig. 7k) further support MC3R's specific role in the regulation of autophagy.

Reviewer #2 (Remarks to the Author):

The addition of new calorimetry data are a nice addition to the overall manuscript and necessary to understand the impact of Mc3r on energy homeostasis

Response: *We thank the reviewer for the positive feedback and appreciate your insights. We hope that the additional data we have now submitted have further strengthened this paper.*

We hope that with this additional work and revised manuscript, our manuscript can now be considered acceptable for publication.

We have revised the paper to address editorial requests and any remaining comments from reviewers.

Reviewer #1 (Remarks to the Author):

The authors have satisfactorily addressed all of my previous concerns. Normalized energy expenditure in the liver Mc3r re-expression mice explains the body weight phenotype. The Seahorse and RNAseq data are also helpful.

Response: *We appreciate the reviewer for the positive feedback and appreciate your insights. We thank you, the reviewers and editor, for the opportunity to consider and now accepting the manuscript for publication.*